# Active Prompt Learning with Vision-Language Model Priors

**Hoyoung Kim**  *hoyoung.kim@postech.ac.kr*
*Graduate School of Artificial Intelligence*
*POSTECH*

**Seokhee Jin**  *jin749@postech.ac.kr*
*Graduate School of Artificial Intelligence*
*POSTECH*

**Changhwan Sung**  *changhwan.sung@postech.ac.kr*
*Graduate School of Artificial Intelligence*
*POSTECH*

**Jaechang Kim**  *jaechang@postech.ac.kr*
*Graduate School of Artificial Intelligence*
*POSTECH*

**Jungseul Ok**  *jungseul.ok@postech.ac.kr*
*Graduate School of Artificial Intelligence*
*POSTECH*

**Reviewed on OpenReview:** *https://openreview.net/forum?id=qBeGCzD3Ij*

## Abstract

Vision-language models (VLMs) have demonstrated remarkable zero-shot performance across various classification tasks. Nonetheless, their reliance on hand-crafted text prompts for each task hinders efficient adaptation to new tasks. While prompt learning offers a promising solution, most studies focus on maximizing the utilization of given few-shot labeled datasets, often overlooking the potential of careful data selection strategies, which enable higher accuracy with fewer labeled data. This motivates us to study a budget-efficient active prompt learning framework. Specifically, we introduce a class-guided clustering that leverages the pre-trained image and text encoders of VLMs, thereby enabling our cluster-balanced acquisition function from the initial round of active learning. Furthermore, considering the substantial class-wise variance in confidence exhibited by VLMs, we propose a budget-saving selective querying based on adaptive class-wise thresholds. Extensive experiments in active learning scenarios across seven datasets demonstrate that our method outperforms existing baselines.

## 1 Introduction

Vision-language models (VLMs), such as CLIP (Radford et al., 2021) and ALIGN (Jia et al., 2021), have demonstrated impressive zero-shot capabilities across various downstream tasks, including object detection (Du et al., 2022; Feng et al., 2022; Zhong et al., 2022), semantic segmentation (Yi et al., 2023; Ghiasi et al., 2022; Li et al., 2022), and image classification (Radford et al., 2021; Singh et al., 2022; Zhai et al., 2022), by aligning visual and textual information within a shared representation space (Radford et al., 2021; Jia et al., 2021; Yuan et al., 2021). Nevertheless, as VLMs rely on manually crafted text prompts for each task, which can be time-consuming and labor-intensive, efficiently adapting VLMs to new tasks remains crucial. Prompt learning has emerged as a promising solution, particularly for image classification tasks, allowing VLMs to learn task-specific prompts without the computational burden of directly fine-tuning image and text encoders.

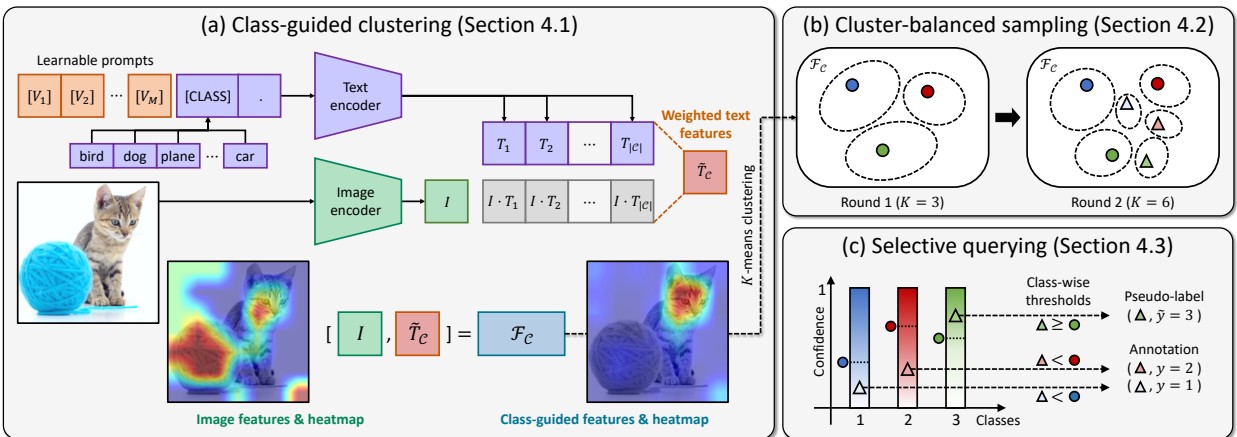

Figure 1: *An overview of the proposed framework.* (a) Class-guided features $\mathcal{F}_\mathcal{C}$ are obtained by concatenating the image features $I$ with the weighted text features $\tilde{T}_\mathcal{C}$, using similarity scores as weights. In the heatmaps, $\mathcal{F}_\mathcal{C}$ focus on the guided-classes $\mathcal{C} = \{\text{Cat}, \text{Dog}\}$ than $I$. (b) $K$-means clustering is performed on $\mathcal{F}_\mathcal{C}$. With an increasing $K$, cluster-balanced sampling becomes available in each round. (c) The confidence scores of previously labeled data (circles) serve as thresholds for new candidates (triangles). If a candidate's confidence exceeds its corresponding threshold, a pseudo-label is assigned to conserve the budget, otherwise it is labeled by annotators.

Generally, prompt learning methods have focused on model-centric approaches, modifying prompt architectures and learning objectives. Specifically, researchers have introduced various prompt types, including text prompts (Zhou et al., 2022b), image-conditioned text prompts (Zhou et al., 2022a) for text encoders, and multimodal prompts that work across both image and text encoders (Khattak et al., 2023a). In terms of learning objectives, prompts are initially trained with cross-entropy loss (Zhou et al., 2022b), supplemented by regularization terms to maintain CLIP's general knowledge (Zhu et al., 2023) and incorporate task-agnostic knowledge (Park et al., 2024), which helps prevent overfitting on specific tasks. However, these model-centric methods mainly focus on leveraging VLM priors to optimize prompts on given few-shot labeled datasets, often overlooking the potential of data selection to achieve higher accuracy with fewer labeled samples.

In contrast to previous model-centric approaches (Zhou et al., 2022b; Zhu et al., 2023), we adopt a data-centric perspective by explicitly leveraging VLM priors to select informative data. In this context, active learning offers an alternative by prioritizing the labeling of the most informative images with minimal budgets. Recent work on active prompt learning (Bang et al., 2024) highlights that class-balanced data selection is crucial for mitigating the inherent imbalanced knowledge within VLMs. However, this approach merely adheres to the conventional few-shot datasets, disregarding the opportunity to fully utilize VLM priors. In contrast, we fully exploit VLM priors through our proposed class-guided clustering and selective querying methods.

For a class-guided clustering, we leverage the pre-trained image and text encoder of VLMs. In detail, we first derive class-guided features by concatenating two components: (i) image features from the image encoder and (ii) text features computed as a weighted sum of each class's text features, with weights based on similarity scores to the image features. We then apply $K$-means clustering (MacQueen et al., 1967) on these class-guided features to achieve balanced data selection across clusters. While traditional active learning often encounters the cold-start problem due to a lack of reliable methods to evaluate data in the initial round (Mahmood et al., 2021; Chen et al., 2023), our cluster-balanced acquisition function provide the benefit of a warm-start. However, purely diversity-focused acquisitions may inefficiently allocate budgets to data in which VLMs are already confident.

To address this issue, we introduce a budget-saving selective querying based on adaptive class-wise thresholds. Since VLMs often exhibit substantial variance in confidence across different downstream tasks and even among individual classes (Bang et al., 2024), we implement adaptive class-wise thresholds without adding extra hyperparameters. Specifically, we assign pseudo-labels to the data selected by the class-balanced

acquisition when their confidence scores exceed the corresponding threshold. As a result, we can conserve budget in each round rather than exhausting it entirely.

The proposed framework, illustrated in Figure 1, fully leverages VLM priors to enable efficient adaptation across various classification tasks. Beyond simply employing the two encoders of CLIP (Radford et al., 2021) for clustering (Li et al., 2024), we further analyze the advantages of incorporating the weighted text features through visualization tools, such as GradCAM (Selvaraju et al., 2017) in Figure 2 and T-SNE (Van der Maaten & Hinton, 2008) in Figure 4. Extensive experiments in active learning scenarios demonstrate that our cluster-balanced acquisition with selective querying outperforms other baselines.

Our main contributions are summarized as follows:

- We propose a budget-efficient active prompt learning for VLMs, particularly on CLIP, where the class-guided clustering and the selective querying fully leverage VLM priors (Sections 4.1 and 4.3).

- We provide in-depth analyses of the class-guided features and clustering with GradFAM, a variant of GradCAM, and T-SNE, respectively (Figures 2 and 4).

- Experiments demonstrate that our method achieve superior budget efficiency and performance across diverse active learning scenarios (Sections 5.2).

- We explore the potential for extending of our data-centric approach into existing model-centric prompt learning methods (Section 5.3).

## 2 Related Work

**Prompt learning in vision-language models.** To address the inefficiency of fine-tuning all VLM parameters, CoOp (Zhou et al., 2022b) proposes prompt learning focused on compact prompts for efficient adaptation. Subsequent works (Zhou et al., 2022a; Khattak et al., 2023a;b; Park et al., 2024; Li et al., 2023) have further developed CoOp, adopting model-centric approaches such as modifying prompt architectures (Khattak et al., 2023a) and learning objectives (Zhu et al., 2023). For instance, MaPle (Khattak et al., 2023a) incorporates multi-modal prompts that jointly consider both VLM encoders, while ProGrad (Zhu et al., 2023) introduces an auxiliary loss term to maintain general knowledge of CLIP. In contrast, PCB (Bang et al., 2024) efficiently adapts VLMs from a data-centric perspective by employing an active learning with a pseudo class-balanced acquisition function. While PCB's acquisition focuses on mitigating the imbalanced prior knowledge of VLMs, we fully exploit VLM priors across the proposed method.

**Active learning in the era of foundation models.** We are in an era where foundation models, such as CLIP (Radford et al., 2021), SAM (Kirillov et al., 2023), and GPT-4 (OpenAI, 2023), dominate a wide range of downstream tasks. Their impressive generalization capabilities may imply that the role of active learning is becoming less significant. However, recent studies continue leveraging the prior knowledge embedded in foundation models (Kim et al., 2024; Gupte et al., 2024; Bayer & Reuter, 2024; Wan et al., 2023) to further enhance budget efficiency with active learning. For instance, ALC (Kim et al., 2024) introduces correction queries to refine SAM-generated labels in semantic segmentation tasks, while ActiveLLM (Bayer & Reuter, 2024) addresses the cold start problem by utilizing LLMs in text classification tasks. We fully leverage VLM priors to enhance budget efficiency in active learning for image classification tasks.

**Acquisition functions in active learning.** Active learning employs various acquisition functions to identify the most informative samples for annotation, aiming to maximize model performance within a constrained budget. These acquisitions are broadly categorized into those focused on uncertainty (Asghar et al., 2017; He et al., 2019; Ostapuk et al., 2019; Fuchsgruber et al., 2024), diversity (Sener & Savarese, 2018; Sinha et al., 2019; Yehuda et al., 2022), and both (Ash et al., 2020a; Hwang et al., 2022; Kim et al., 2023; Wang et al., 2019; Hwang et al., 2023). Recent studies show that uncertainty-based acquisitions are more effective with higher budgets, while diversity-based ones perform better with lower budgets (Hacohen et al., 2022; Hacohen & Weinshall, 2023a;b). Building on this, we propose a cluster-balanced acquisition to adapt VLMs within limited budgets. In addition, we introduce a selective querying with adaptive class-wise thresholds to further conserve budget.

---

**Algorithm 1** Proposed Active Prompt Learning

---

**Require:** Image set $\mathcal{I}$, initial prompts $t_0$, budget per round $B$, and the number of rounds $R$
1: **for** $r = 1, 2, \ldots, R$ **do**
2:     Extract class-guided features $\forall i \in \mathcal{I}$ by combining image and weighted text features via (7)
3:     Perform K-means clustering on class-guided features
4:     Select representative images from each cluster as candidates via (9)
5:     Compute class-wise confidence thresholds using previously labeled data via (11)
6:     Construct dataset $\mathcal{D}_r$ by assigning pseudo or ground-truth labels via selective querying in (12)
7:     Reinitialize and train prompts $t_r$ using dataset $\mathcal{D}_r$ with the objective in (13)
8: **end for**
9: **return** Final dataset $\mathcal{D}_R$ and trained prompts $t_R$

---

## 3 Preliminaries

For efficient adaptation in vision-language models (VLMs), we fully leverage their prior knowledge. Before presenting our method, we first outline the structure of VLMs and the relevant priors, in Section 3.1, which are utilized in our method. After that, we describe the basics of prompt learning with few-shot datasets in Section 3.2.

### 3.1 VLM Priors

Pre-trained VLMs have shown decent zero-shot performance across various classification tasks. Specifically, the CLIP model $\theta$ (Radford et al., 2021) comprises an image encoder $\theta_{\text{img}}$ and a text encoder $\theta_{\text{txt}}$, performing zero-shot inference on an image $x$ for a target class $c \in \mathcal{C}$ using cosine similarity as:

$$p_\theta(y = c \mid x; t, \mathcal{C}) := \frac{\exp\big(\cos(\theta_{\text{img}}(x), \theta_{\text{txt}}(t_c))/\tau\big)}{\sum_{k \in \mathcal{C}} \exp\big(\cos(\theta_{\text{img}}(x), \theta_{\text{txt}}(t_k))/\tau\big)} \, , \tag{1}$$

where the text prompt $t$ can be set as "a photo of a ", and $t_c$ represents the concatenation of $t$ and "[CLASS]." with the class token replaced by the class name $c$. Here, $\tau$ denotes a temperature parameter. Considering the highest probability among all classes, we can obtain the pseudo label of an image $x$ as follows:

$$y_\theta(x; t, \mathcal{C}) := \arg\max_{c \in \mathcal{C}} p_\theta(y = c \mid x; t, \mathcal{C}) \, . \tag{2}$$

We note that the pseudo label depends on the text prompt $t$ and the target class set $\mathcal{C}$. In Section 4, we leverage the pre-trained image and text encoders of CLIP for our class-guided clustering and the pseudo label for our selective querying.

### 3.2 Prompt Learning in VLMs

Vision-language models (VLMs) contain numerous parameters, which makes fine-tuning on a small labeled dataset impractical. Recently, prompt learning has facilitated efficient adaptation in VLMs by freezing the image and text encoders and focusing on learning input prompts. For instance, CoOp (Zhou et al., 2022b) introduces learnable vectors into the text prompt $t_c$ of class $c$, replacing the conventional "a photo of a [CLASS]." text prompts with:

$$t_c := [V_1][V_2] \ldots [V_M][\text{CLASS}]. \, , \tag{3}$$

where each $V$ represents a learnable vector and $M$ denotes the number of vectors. These vectors in $t$ are trained on a dataset $\mathcal{D}$ by minimizing the cross-entropy (CE) loss:

$$\hat{\mathbb{E}}_{(x,y)\sim\mathcal{D}}\big[\text{CE}\big(y, p_\theta(y; x, t, \mathcal{C})\big)\big] \, . \tag{4}$$

In this context, previous prompt learning methods have primarily taken a model-centric perspective, aiming to maximize model performance on the given training dataset $\mathcal{D}$. From a data-centric perspective, however,

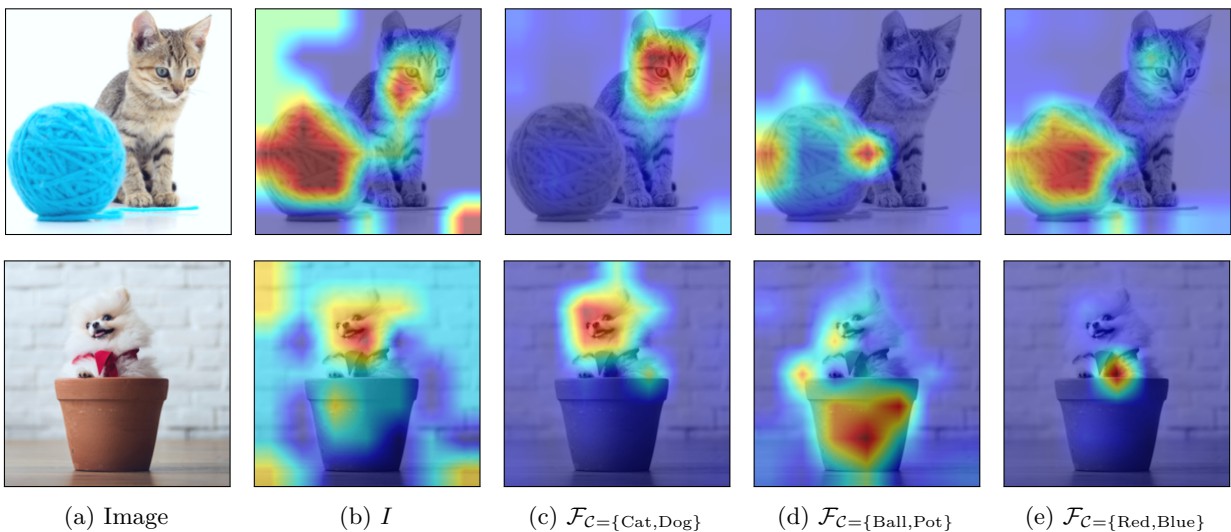

| (a) Image | (b) $I$ | (c) $\mathcal{F}_{\mathcal{C}=\{\text{Cat,Dog}\}}$ | (d) $\mathcal{F}_{\mathcal{C}=\{\text{Ball,Pot}\}}$ | (e) $\mathcal{F}_{\mathcal{C}=\{\text{Red,Blue}\}}$ |

Figure 2: *GradFAM with various target features.* (b) All objects in the image significantly impacts the target image features $I$. (c-e) With our class-guided features $\mathcal{F}_{\mathcal{C}}$ for target features, the heatmap aligns with the target classes $\mathcal{C}$. Further details are in the Appendix A.

building datasets requires human labor, making active learning essential for creating information-dense datasets while minimizing interactions with annotators. In addition, we note that our data-centric approach is compatible with existing model-centric prompt learning methods.

## 4 Proposed Active Prompt Learning

Given an unlabeled image set $\mathcal{U}$, we consider a general active learning scenario for image classification tasks, where annotators are asked to label each image $x \in \mathcal{U}$ with its accurate label $y \in \mathcal{C} := \{1, 2, ..., C\}$. For each round $r$, with a maximum budget of $B$, we begin by constructing a candidate set of $B$ images from $\mathcal{U}_r$ utilizing class-guided clustering. Each candidate is then evaluated with a class-wise threshold to determine whether to acquire ground-truth labels from annotators or use pseudo labels to conserve budget. Finally, learnable prompts $t_r$ are trained by prompt learning on the dataset $\mathcal{D}_r$ accumulated up to round $r$.

In the followings, we introduce a class-guided clustering, which leverages the pre-trained image and text encoders of CLIP (Section 4.1), and a cluster-balanced acquisition function to select candidate images (Section 4.2). After that, we introduce a budget-saving selective querying (Section 4.3). The overall algorithm is described in Algorithm 1.

### 4.1 Class-Guided Clustering

Conventional active learning often relies on random sampling to construct the initial dataset, which can lead to the cold-start problem. In contrast, we fully utilize CLIP's pre-trained image and text encoders for class-guided clustering, which can be applied at each round of the active learning process. To construct class-guided features of an image $x$, we first define its image and text features, $I(x)$ and $\tilde{T}_{\mathcal{C}}(x)$, respectively. For the image features, we simply use the image encoder of CLIP as follows:

$$I(x) := \theta_{\text{img}}(x) . \tag{5}$$

To derive the text features of $x$, we compute a weighted combination of text features with soft labels as weights:

$$\tilde{T}_{\mathcal{C}}(x; t_{r-1}) := \sum_{c \in \mathcal{C}} p_\theta(y = c \mid x, t_{r-1}) \theta_{\text{txt}}(t_{r-1,c}) , \tag{6}$$

where $t_{r-1}$ represents the learned vectors from the previous round, and $t_{r-1,c}$ concatenates $t_{r-1}$ with class $c$ as described in (3). For the initial round, we set $t_{r-1}$ as "a photo of a ". Finally, we concatenate the image and text features to obtain the class-guided features $\mathcal{F}_{\mathcal{C}}(x)$ for image $x$ as:

$$\mathcal{F}_{\mathcal{C}}(x; t_{r-1}) := [I(x), \tilde{T}_{\mathcal{C}}(x; t_{r-1})] . \tag{7}$$

After that, we apply the $K$-means clustering algorithm (MacQueen et al., 1967) on the set of class-guided features across all images to partition them into $K$ clusters.

**GradFAM on class-guided features.** We modify GradCAM (Selvaraju et al., 2017) into GradFAM (Gradient-weighted Feature Activation Mapping) to visualize the influence of class-guided features on an image. GradCAM highlights the degree to which individual pixels contribute to a specific class by analyzing the gradients of the class score with respect to the feature maps. To tailor GradCAM for VLMs, we introduce the concept of *target features* denoted by $\mathcal{F}_{\text{target}}$, allowing GradCAM to visualize the influence of target features on an image $x$ based on the cosine similarity score defined as $\cos(I(x), \mathcal{F}_{\text{target}})$. Our analysis technique, called GradFAM, visualizes the influence of target features rather than target class, offering the advantage of enabling label-free analysis. Figure 2b represents the case where $\mathcal{F}_{\text{target}} = I(x)$. Consistent with CLIP's approach of embedding images and texts into a shared space, we observe that the influence of the target image features primarily concentrates on the overall objects within the image. On the other hand, when the target features are guided by the set of classes, *i.e.* $\mathcal{F}_{\text{target}} = \mathcal{F}_{\mathcal{C}}(x; t)$ where $t$ is "a photo of a ", Figures 2c to 2e distinctly highlight the specific objects corresponding to the guiding class sets. To compute cosine similarity, we concatenate two copies of the same image features to match the dimensionality of the other features. We note that our class-guided features incorporates class-relevant information through the text encoder, potentially resulting in clustering that aligns more closely with the target classifier, as described in Figure 4.

## 4.2 Cluster-Balanced Acquisition Function

For ease of explanation, we first outline the data selection process in the initial round of active learning. For a cluster-balanced acquisition, we set the number of clusters $K$ equal to the budget $B$, allowing for the selection of one image per cluster. To choose the most representative sample from each cluster $C_i$, we first calculate its centroid $c_i$ as follows:

$$c_i := \frac{1}{|C_i|} \sum_{x \in C_i} \mathcal{F}_{\mathcal{C}}(x; t) . \tag{8}$$

The closest image $x_i^*$ to the centroid $c_i$ is then selected as:

$$x_i^* := \arg\min_{x \in C_i} ||\mathcal{F}_{\mathcal{C}}(x; t) - c_i||_2 . \tag{9}$$

We can construct the set of candidate images for querying:

$$\mathcal{Q} := \{x_1^*, x_2^*, \ldots, x_B^*\} . \tag{10}$$

In the initial round, we consume the entire budget $B$ to request annotations for all candidate images in $\mathcal{Q}$. Here, $t$ is replaced by $t_{r-1}$, and $\mathcal{Q}$ by $\mathcal{Q}_r$ for a subsequent round $r$.

**Subsequent rounds with increasing $K$.** To enhance the diversity in the selected data, we introduce a progressively increasing $K$ based on round $r$, *i.e.* $K = B \times r$. This linear increase in $K$ ensures that at least $B$ clusters remain unlabeled in each round, allowing for the selection of clusters not included in previous rounds. However, due to the inaccuracy of clustering in earlier rounds, samples previously assigned to different clusters may now be classified into the same cluster. To address these cases, we prioritize larger clusters by sorting all unlabeled clusters by size and selecting the top-$K$ clusters. Ablation studies on $K$ are in the Appendix C.

### 4.3 Selective Querying

In each round $r$, we can allocate the entire budget $B$ to acquire labels for all candidate images in $\mathcal{Q}_r$. However, CLIP has demonstrated decent zero-shot performance in downstream classification tasks (Radford et al., 2021) and performs even better with a few labeled samples (Zhou et al., 2022b). Therefore, for a candidate image where CLIP is already sufficiently confident in its label, we skip manual labeling and apply a pseudo label to conserve budget.

Since CLIP's knowledge is imbalanced across classes in classification tasks (Bang et al., 2024), we propose a selective querying with class-wise thresholds. To this end, we leverage the confidence scores of images from the previous training dataset $\mathcal{D}_{r-1}$. In round $r$, the threshold $\epsilon_{r,c}$ for class $c \in \mathcal{C}$ is computed as follows:

$$\epsilon_{r,c} := \frac{1}{|\mathcal{D}_{r-1,c}|} \sum_{(x,y=c)\in\mathcal{D}_{r-1,c}} p_\theta(y \mid x; t_{r-1}, \mathcal{C}) \, , \tag{11}$$

where $\mathcal{D}_{r-1,c}$ represents the subset of the training dataset labeled as $c$. Note that thresholds for round $r$ depend on prior information, including $\mathcal{D}_{r-1}$ and $t_{r-1}$. Therefore, a selective querying is impossible in the initial round. We finally apply pseudo labels to candidates if their confidence exceeds the corresponding threshold. As a result, the dataset $\mathcal{D}_r$ at round $r$ can be constructed as follows:

$$\begin{aligned} \mathcal{D}_r :=& \{(x,y) \mid x \in \mathcal{Q}_r, \ p_\theta(\tilde{y} \mid x; t_{r-1}, \mathcal{C}) < \epsilon_{r,\tilde{y}}\} \cup \mathcal{D}_{r-1} \\ & \cup \{(x,\tilde{y}) \mid x \in \mathcal{Q}_r, \ p_\theta(\tilde{y} \mid x; t_{r-1}, \mathcal{C}) \geq \epsilon_{r,\tilde{y}}\} \, , \end{aligned} \tag{12}$$

where the pseudo label $\tilde{y} = y_\theta(x; t, \mathcal{C})$ is defined as in (2). Here, we note that $|\mathcal{D}_r| = B \times r$, yet the budget required is actually lower thanks to our selective querying.

To avoid bias in prompts trained during the previous round, we reinitialize $t_r$ randomly at each round and train prompts by minimizing the following CE loss:

$$\hat{\mathbb{E}}_{(x,y)\sim\mathcal{D}_r}\big[\mathrm{CE}\big(y, p_\theta(y; x, t_r, \mathcal{C})\big)\big] \, . \tag{13}$$

**Revisiting a unified prompt for prompt learning.** Recent studies in prompt learning for VLMs have introduced sophisticated prompt designs, including image-wise (Zhou et al., 2022a; Yao et al., 2024) and class-wise prompts (Zhou et al., 2022b; Bang et al., 2024; Yao et al., 2024). However, these prompts are prone to overfitting, particularly in active learning scenarios where only a limited number of samples are available. To mitigate this issue, we introduce a new similarity measure that incorporates both a unified prompt $t_u$ and class-wise prompts $t_\mathcal{C} = \{t_c \mid c \in \mathcal{C}\}$. Specifically, we define the cosine similarity between image $x$ and class $c$ as:

$$\frac{\cos(\theta_{\mathrm{img}}(x), \theta_{\mathrm{txt}}(t_{u,c})) + \cos(\theta_{\mathrm{img}}(x), \theta_{\mathrm{txt}}(t_c))}{2} \, , \tag{14}$$

where $t_{u,c}$ concatenates $t_u$ with class $c$. We demonstrate that incorporating a unified prompt is beneficial in active learning scenarios with limited resources and enhances the effectiveness of the proposed selective querying.

## 5 Experiments

### 5.1 Experimental Setup

**Datasets and implementation details.** Following a previous study (Bang et al., 2024), we use seven publicly available image classification datasets: OxfordPets (pet species) (Parkhi et al., 2012), FGVCAircraft (aircraft types) (Maji et al., 2013), Caltech101 (general object categories) (Fei-Fei et al., 2004), Flowers102 (flower species) (Nilsback & Zisserman, 2008), DTD (texture patterns) (Cimpoi et al., 2014), StanfordCars (car models) (Krause et al., 2013), and EuroSAT (satellite land cover types) (Helber et al., 2019). In our experiments, we employ CLIP ViT-B/32 (Dosovitskiy et al., 2021; Radford et al., 2021) as our VLM. At each

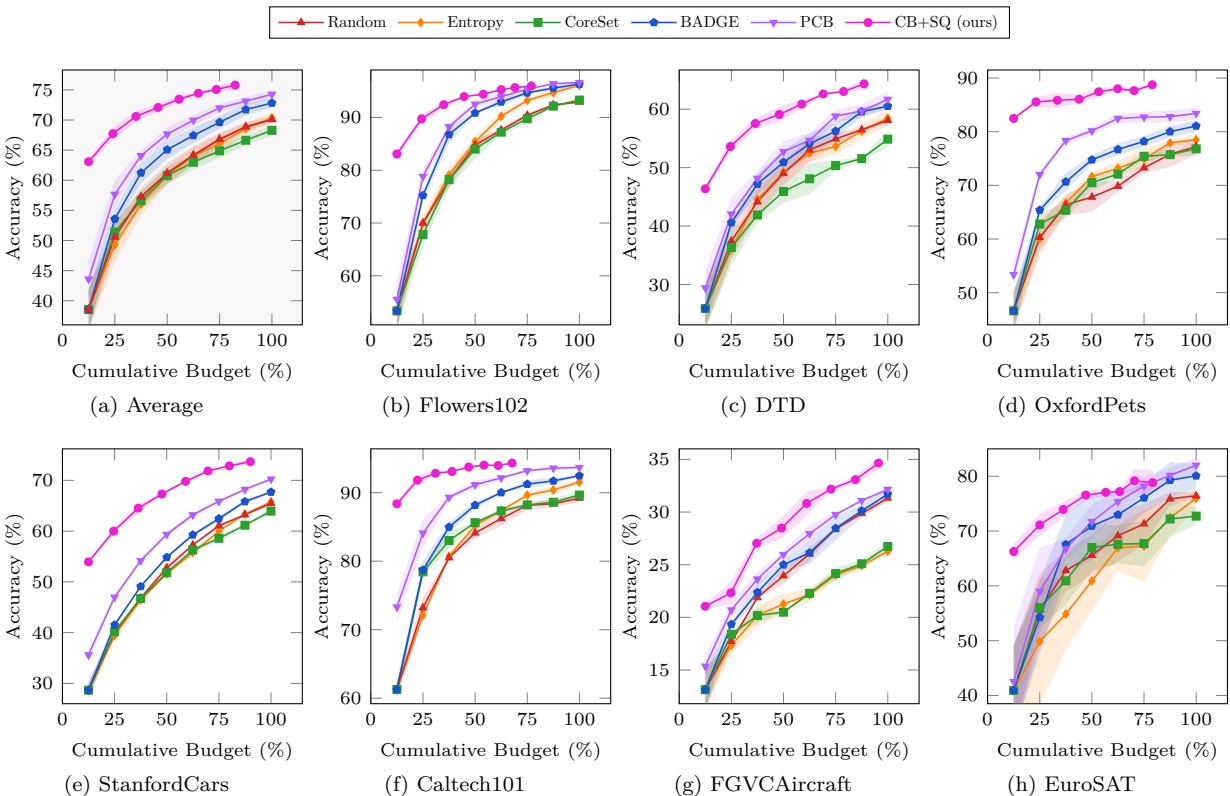

Figure 3: *Effect of the proposed method.* (a-h) Our CB+SQ outperforms the other baselines in average performance across 7 datasets. Experiments with a comparable budget are provided in Figure 11.

round $r$, we reinitialize the learnable prompts $t_r$, consisting of 16 vectors, using a Gaussian distribution with a mean of 0 and a standard deviation of 0.02. Following the training details in CoOp (Zhou et al., 2022b), we train these prompts for 200 epochs per round using the SGD optimizer, initialized with a learning rate of 0.002 and decaying according to a cosine annealing.

## 5.2 Active Learning Scenario

**Baselines.** Our cluster-balanced acquisition with selective querying (CB+SQ) is compared with the state-of-the-art (SOTA) active prompt learning method for VLMs, known as pseudo-class balance (PCB) (Bang et al., 2024), which operates based on BADGE (Ash et al., 2020b). In addition, we compare with conventional acquisitions commonly used in active learning for classification tasks, including Random, Entropy Holub et al. (2008) and CoreSet (Sener & Savarese, 2018). To ensure a fair reproduction of all baseline results, we follow the experimental setting of PCB, leveraging class descriptions from LLMs (Menon & Vondrick, 2022) to train class-wise text prompts following CoOp (Zhou et al., 2022b).

**Evaluation protocol.** For a fair comparison, we follow the active learning scenario established in PCB (Bang et al., 2024). Specifically, experiments are conducted over 8 rounds, with the maximum budget per round set to the number of classes, *i.e.* $B = |\mathcal{C}|$. Since the total budget varies across datasets, we set it to be fully spent at 100% by the 8th round, with a maximum budget of 12.5% per round. Here, a budget of one indicates that an oracle assigns the ground-truth label for a single image. Thanks to our selective querying in CB+SQ, we generally consume less budget per round than other baselines from the second round onward. We report the average accuracy over three trials, with shaded regions in the graphs representing the standard deviation.

**Effect of proposed method.** Figure 3 demonstrates the effectiveness of our proposed acquisition CB+SQ across seven datasets. While other baselines rely on Random acquisition to mitigate the cold-start problem, *i.e.* initial performance degradation compared to Random, our CB+SQ leverages the pre-trained image and

Table 1: *Synergy of the proposed acquisition with existing prompt learning methods.* All baseline methods are trained with 1-shot datasets. Our CB*-based curated datasets enhance the performance of previous model-centric prompt learning methods.

| Methods | Flowers102 | DTD | OxfordPets | StanfordCars | Caltech101 | Aircraft | EuroSAT | Average (%) |
|---|---|---|---|---|---|---|---|---|
| MaPle | 75.23 | 48.77 | 85.27 | 57.30 | 91.57 | 18.33 | 61.67 | 62.59 |
| + CB | 78.80 | 46.47 | 87.23 | 55.70 | 90.17 | 16.00 | **73.50** | 63.98 |
| + CB* | **80.70** | **51.07** | **88.83** | **60.47** | **93.07** | **20.85** | 71.07 | **66.58** |
| PromptSRC | 77.60 | 51.53 | 89.60 | 63.67 | 93.10 | 18.67 | 66.93 | 65.87 |
| + CB | 79.73 | 50.87 | 89.80 | 61.30 | 92.40 | **22.03** | 70.20 | 66.62 |
| + CB* | **80.43** | **56.17** | **90.13** | **63.97** | **93.97** | 21.27 | **70.33** | **68.04** |
| ProMetaR | 78.87 | 52.07 | 87.63 | 60.60 | 92.53 | 19.50 | 68.13 | 65.62 |
| + CB | 80.43 | 49.10 | 89.53 | 59.37 | 91.80 | 20.37 | 71.80 | 66.06 |
| + CB* | **83.37** | **55.20** | **89.53** | **63.83** | **93.90** | **22.10** | **72.87** | **68.69** |

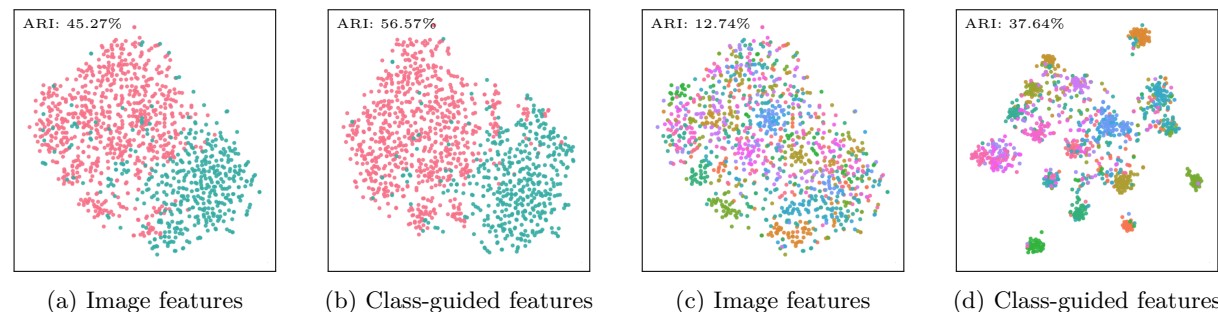

(a) Image features     (b) Class-guided features     (c) Image features     (d) Class-guided features

Figure 4: *T-SNE for class-guided clustering.* (a, c) Clustering based solely on image features results in clusters that are poorly separated. (a) and (c) correspond to the $|\mathcal{C}| = 2$ and $|\mathcal{C}| = 20$ settings, respectively. (b, d) In contrast, our class-guided clustering leads to more distinct clusters that align with the size of the guiding class set $\mathcal{C}$.

text encoders of CLIP to enable a warm-start, leading to consistently strong early performance. As shown in Figure 3a, our method shows a 19.5%p performance gain over the baselines at the first acquisition round. Notably, with only $|\mathcal{C}|$ queried samples, our CB+SQ already outperforms other baselines trained with $3|\mathcal{C}|$ samples, highlighting its sample efficiency. In addition, our selective querying allows CB+SQ to outperform other baselines while reducing the labeling budget by 17.6%. These gains are consistent across both fine-grained and general datasets, and we also observe the same phenomenon reported in Bang et al. (2024), where conventional acquisition methods such as Entropy and CoreSet underperform even Random sampling.

### 5.3 Further Analyses

**Extensions to SOTA model-centric prompt learning methods.** Our cluster-balanced (CB) acquisition effectively selects informative images by utilizing a pre-trained CLIP model. These selected images can be directly applied to enhance the performance of SOTA model-centric prompt learning methods, including MaPle (Khattak et al., 2023a), PromptSRC (Khattak et al., 2023b), and ProMetaR (Park et al., 2024). Table 1 shows that our CB-based datasets slightly outperforms the conventional 1-shot datasets, which contain one labeled image per class. More specifically, following the few-shot setting of CoOp (Zhou et al., 2022b), we assume access to ground-truth labels to construct a perfectly class-balanced 1-shot dataset by randomly selecting one sample from each class. However, the 1-shot datasets differ from active learning, which start with unlabeled images. For a fair comparison with them, we construct CB*-based datasets, where the weighted text features in (6) are replaced with the text features of the ground-truth label. In Table 1, our CB*-based datasets outperform other baselines, emphasizing the importance of data-centric approaches for efficient adaptation in VLMs.

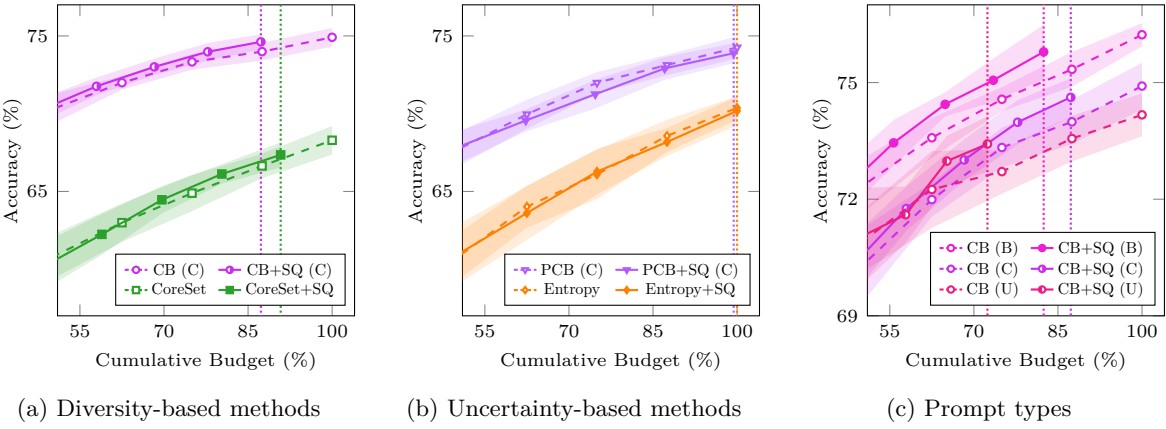

(a) Diversity-based methods   (b) Uncertainty-based methods   (c) Prompt types

Figure 5: *Ablation studies on selective querying.* Each vertical dotted line indicates the effect of selective querying (SQ). U, C, and B denote unified prompt, class-wise prompts, and both prompts. SQ is more effective for (a) diversity-based acquisitions, such as our CB and CoreSet, compared to (b) uncertainty-based methods. (c) The benefit of SQ becomes more pronounced when unified prompts (U) are used.

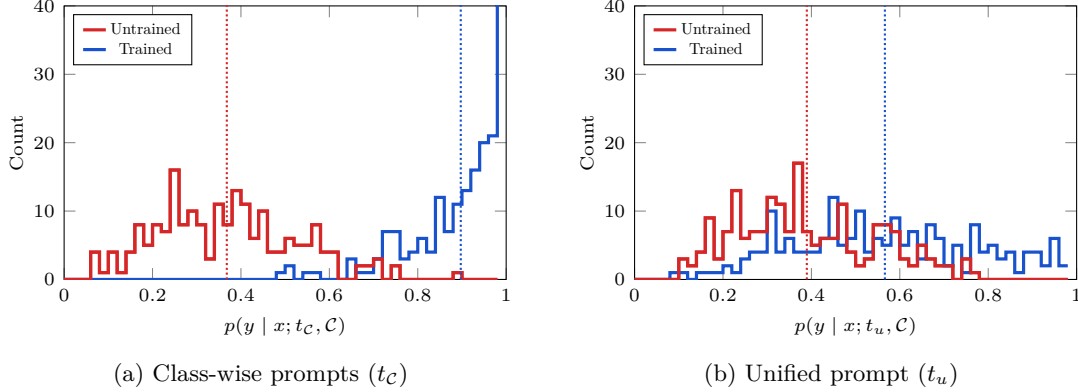

(a) Class-wise prompts ($t_\mathcal{C}$)   (b) Unified prompt ($t_u$)

Figure 6: *Confidence distributions.* Due to overfitting in (a) the class-wise prompts, the overall confidence scores and the mean confidence (dotted line) of trained samples are significantly higher than those obtained with (b) the unified prompt.

**Visualization for class-guided clustering.** In Section 4.1, we analyze the proposed class-guided features using GradFAM, as shown in Figure 2. Here, we further visualize the impact of class-guided features on clustering with T-SNE (Van der Maaten & Hinton, 2008) on the Waterbirds dataset (Sagawa et al., 2019). Specifically, the Waterbirds dataset consists of 200 distinct bird species, where each image is labeled by habitat (water, land), background (water, land), and specific species. This allows us to categorize the dataset into 2, 4, or 200 groups. For improved visual clarity, we analyze a subsample of 20 classes. Figures 4a and 4c shows that the limitation of relying solely on image features in capturing semantic information. In contrast, our class-guided features, which incorporate the set of classes $\mathcal{C}$, effectively reflect relevant class information in the clustering results, as illustrated in Figures 4b and 4d. This improvement is also reflected in the Adjusted Rand Index (ARI), where the score increases from 12.74% to 37.64% when moving from image-only features to class-guided features. More details are provided in Appendix C.

**Synergy of selective querying with our method.** Our selective querying (SQ) synergizes strongly with our method for two key reasons. First, our cluster-balanced acquisition follows a diversity-driven strategy, which may occasionally select well-understood samples. In such cases, SQ leverages pseudo labels to conserve budget. As shown in Figure 5a, SQ is particularly effective in diversity-based methods such as CoreSet and CB, while uncertainty-based approaches like Entropy and PCB show limited improvement in Figure 5b. Second, we

Table 2: *Ablation studies on different feature spaces.* In the initial round, our class-guided features (third row), leveraging both image and text features, demonstrate effectiveness across 7 datasets.

| Image Features | Text Features | Ground-truth | Average Acc. (%) |
|:---:|:---:|:---:|:---:|
| ✓ | ✗ | ✗ | $52.72_{\pm 0.37}$ |
| ✗ | ✓$_{\text{soft}}$ | ✗ | $48.24_{\pm 0.21}$ |
| ✓ | ✓$_{\text{soft}}$ | ✗ | $\mathbf{55.92}_{\pm 0.38}$ |
| ✗ | ✗ | ✓ | $48.10_{\pm 0.41}$ |
| ✓ | ✓$_{\text{label}}$ | ✓ | $\mathbf{58.27}_{\pm 0.43}$ |

incorporate a unified prompt, as described in Section 4.3, further enhancing the effectiveness of SQ. Figure 6 shows that the unified prompt results in confidence scores for trained samples with less overfitting compared to class-wise prompts, leading to more reliable class-wise thresholds in (11). This observation aligns with the results in Figure 5c, showing that a unified prompt is more effective in reducing the labeling budget than class-wise prompts.

**Confidence distributions by prompt type.** Figure 6 compares the confidence distributions of trained and untrained samples. We first train either class-wise prompts $t_{\mathcal{C}}$ or a unified prompt $t_u$ on the StanfordCars dataset $\mathcal{D}$ (Krause et al., 2013). To emulate an active-learning setting with a limited budget, we form a 4-shot training subset $\mathcal{D}_t \subset \mathcal{D}$. For each class $c$, we then compute the average confidence score $s_{t,c}$ as follows:

$$s_{t,c} := \frac{1}{|\mathcal{D}_{t,c}|} \sum_{(x,y) \in \mathcal{D}_{t,c}} p(y \mid x; t, \mathcal{C}) \,, \tag{15}$$

where $\mathcal{D}_{t,c}$ is the portion of $\mathcal{D}_t$ that belongs to class $c$, and $t$ is either $t_{\mathcal{C}}$ or $t_u$. We then plot 50-bin histograms of these scores, using blue for the training subset $\mathcal{D}_t$ and red for the remaining data $\mathcal{D} \setminus \mathcal{D}_t$. The resulting plots show that class-wise prompts concentrate confidence on seen samples, which may indicate overfitting, whereas the unified prompt yields a more balanced confidence distribution across unseen data.

**Effect of various feature spaces.** In Table 2, we examine the effect of various feature spaces on our cluster-balanced acquisition, as an alternative to the proposed class-guided features that leverage both Image and Text features. Table 2 demonstrates that our class-guided features yield a 3.2%p and 7.7%p improvement over using only Image and only Text features, respectively. In addition, we experiment with a conventional few-shot labeled dataset that relies solely on Labels for a perfectly class-balanced dataset, which can be impractical in an active learning scenario. For a fair comparison, we redefine the weighted text features in (6) as the text features of ground-truth labels, resulting in an average performance increase of 10.2%p.

**Analyses of constructed datasets.** A primary goal of active learning is to construct high-quality datasets with minimal labeling budgets. Figure 7 demonstrates that our method builds high-quality datasets with a significantly smaller budget compared to previous methods. Specifically, while previous methods consume the entire budget each round to generate a 100% clean dataset, our selective querying conserves budget by using pseudo-labels based on class-wise thresholds, yet still achieves a comparable level of dataset quality. This results in performance on par with previous methods, as shown in Figure 3. Additionally, we note that as rounds progress, the budget-saving advantage of our method becomes more pronounced.

**ImageNet experiments.** ImageNet (Deng et al., 2009) contains 1.28 M training images, making large-scale active learning computationally demanding. Emam et al. (2021) reports that CoreSet and BADGE are infeasible at this scale, and that PCB, which builds on BADGE, suffers the same limitation. Our method overcomes this bottleneck by employing a lightweight $K$-means clustering step. As shown in Figure 8, the resulting CB+SQ strategy scales efficiently to ImageNet and achieves higher accuracy than all competing baselines.

**Base-to-novel generalization.** Following previous work (Zhou et al., 2022b), we divide each dataset's classes into base and novel groups. We then use acquisition functions to select a subset of unlabeled images

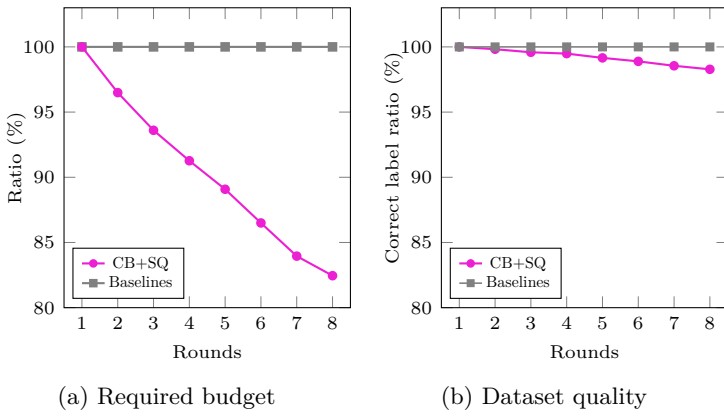

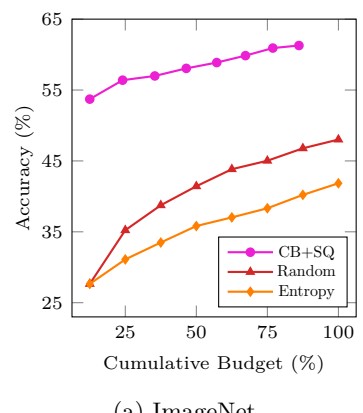

(a) Required budget      (b) Dataset quality      (a) ImageNet

Figure 7: *Analyses of constructed datasets.* (a) As rounds progress, we construct datasets with progressively smaller budgets. (b) The quality of the constructed dataset remains consistently high, regardless of the number of rounds.

Figure 8: *ImageNet experiments.* Our approach is applicable to ImageNet and outperforms existing methods.

Table 3: *Effectiveness on unseen classes.* Our CB shows strong performance on novel classes in the initial round across 9 datasets.

| Methods | Base | Novel | HM |
|---------|------|-------|-----|
| Random | $59.02_{\pm1.06}$ | $58.94_{\pm1.48}$ | $58.98_{\pm1.14}$ |
| CB | $\mathbf{68.81}_{\pm1.07}$ | $\mathbf{63.68}_{\pm1.25}$ | $\mathbf{66.15}_{\pm0.67}$ |

from the base classes to train prompts, which are subsequently applied to the novel classes for evaluation. In Table 3, we compare our cluster-balanced (CB) acquisition with Random reflecting the cold-start of other acquisitions in Figure 3. Table 3 demonstrates that our CB acquisition outperforms Random in both base and novel groups. Here, HM denotes the harmonic mean of base and novel performance.

**Various ablation studies.** Figure 9a examines the effect of different prompt types, including unified (U), class-wise (C), and both (B), revealing that our CB consistently outperforms PCB regardless of the prompt type. In addition, we analyze the contribution of the proposed components: (i) selective querying (SQ) and (ii) the unified prompt (C to B), as illustrated in Figures 9b and 9c. Although CoreSet has demonstrated low performance in active learning for VLMs (Bang et al., 2024), this is primarily due to its reliance solely on image features. We observe that incorporating our class-guided features can enhance its performance to some extent as shown in Figure 9c, though it still fails to surpass PCB. We note that active learning for conventional neural networks with image-only encoders may differ from active learning for VLMs.

## 6 Conclusion

In this work, we propose an active prompt learning framework that leverages the prior knowledge of vision-language models (VLMs) for efficient data acquisition and prompt adaptation. Using the pre-trained image and text encoders of CLIP, we extract class-guided features that combine image embeddings with weighted text embeddings based on similarity scores, enabling K-means clustering to select diverse and representative samples. To further enhance budget efficiency, we introduce a selective querying strategy with adaptive class-wise thresholds, assigning pseudo-labels to high-confidence samples while requesting annotations for uncertain ones. Experiments across seven datasets and large-scale settings such as ImageNet show that our cluster-balanced acquisition with selective querying reduces annotation costs and achieves superior accuracy compared to baselines. Furthermore, our data-centric approach complements existing model-centric prompt learning methods, offering a general strategy for scalable VLM adaptation.

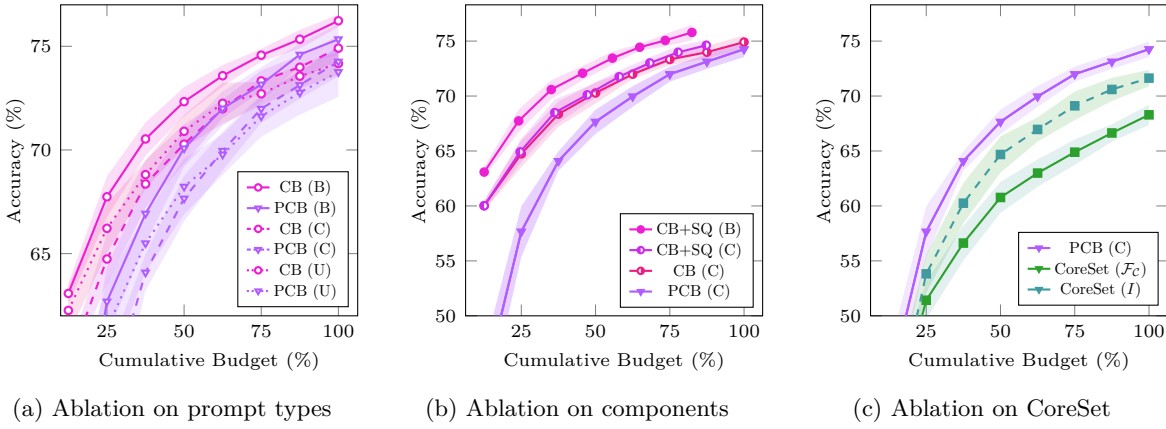

(a) Ablation on prompt types  (b) Ablation on components  (c) Ablation on CoreSet

Figure 9: *Various ablation studies.* (a) Our CB outperforms PCB regardless of whether unified prompt (U), class-wise prompts (C), or both (B) are used. (b) Even with one component removed, our method still outperforms PCB. (c) The performance of CoreSet improves with our class-guided features.

**Limitations and future work.** Our framework relies on strong pre-trained vision-language models such as CLIP, which may limit its effectiveness when applied to weaker or domain-specific backbones. Moreover, it has so far been validated only on image classification tasks, leaving its applicability to other vision tasks. Future work will explore extending the approach to object detection, semantic segmentation, and real human-in-the-loop settings to assess its broader utility.

**Acknowledgements.** This work was partly supported by the IITP grants and the NRF grants funded by Ministry of Science and ICT, Korea (No.RS-2019-II191906, Artificial Intelligence Graduate School Program (POSTECH); No.RS-2021-II212068, Artificial Intelligence Innovation Hub; No.RS-2023-00217286; No.RS-2021-II210739, Development of Distributed/Cooperative AI based 5G+ Network Data Analytics Functions and Control Technology; No.RS-2024-00457882, AI Research Hub Project).

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

# A   Details of GradFAM

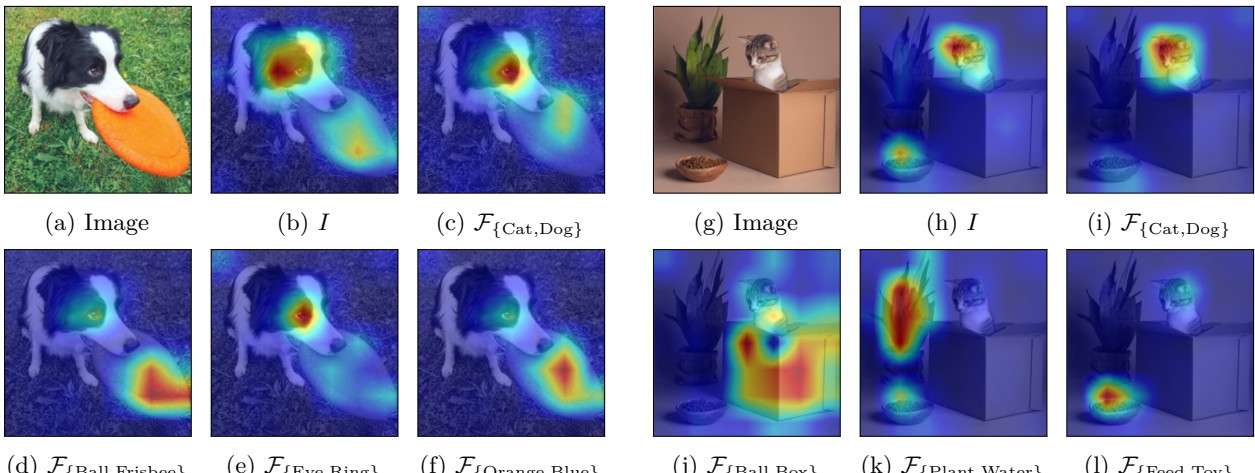

| (a) Image | (b) $I$ | (c) $\mathcal{F}_{\{\text{Cat,Dog}\}}$ | (g) Image | (h) $I$ | (i) $\mathcal{F}_{\{\text{Cat,Dog}\}}$ |

| (d) $\mathcal{F}_{\{\text{Ball,Frisbee}\}}$ | (e) $\mathcal{F}_{\{\text{Eye,Ring}\}}$ | (f) $\mathcal{F}_{\{\text{Orange,Blue}\}}$ | (j) $\mathcal{F}_{\{\text{Ball,Box}\}}$ | (k) $\mathcal{F}_{\{\text{Plant,Water}\}}$ | (l) $\mathcal{F}_{\{\text{Feed,Toy}\}}$ |

Figure 10: GradFAM visualizations for two image examples under diverse class-guided feature targets.

In Section 4.1, we propose class-guided features and analyze them with our GradFAM, a modified version of GradCAM (Selvaraju et al., 2017). Here, we provide a detailed description of GradFAM and its differences from GradCAM.

**Gradient-weighted class activation map (CAM).** To highlight the importance of regions in an image $x$ associated with a given class $c$, Grad-CAM introduces the class-discriminative localization map as follows:

$$L^c_{\text{CAM}}(x) \in \mathbb{R}^{U \times V} \; , \tag{16}$$

where $U$ and $V$ are the width and height of the image $x$.

Let $f(x;\theta) \in \mathbb{R}^{|\mathcal{C}|}$ represent the output logits of a neural network with parameters $\theta$, where $|\mathcal{C}|$ is the total number of classes. For a given class $c \in \mathcal{C}$, the score $y^c(x;\theta)$ is defined as follows:

$$y^c(x;\theta) := f_c(x;\theta) \; , \tag{17}$$

where $f_c(x;\theta)$ denotes the $c$-th logit value, representing the model's confidence for class $c$ given the image $x$. Grad-CAM computes the gradient of $y^c(x;\theta)$ with respect to the activation maps $A^k$ in the last convolutional layer, where $A^k \in \mathbb{R}^{W \times H}$ represents the activation map of the $k$-th channel with spatial dimensions $W$ and $H$. Note that $W$ and $H$ are typically smaller than $U$ and $V$ due to downsampling in the convolutional layers.

The importance weight $\alpha^k_c$ for each channel $k$ is obtained by applying global average pooling over the gradients as:

$$\alpha^{c,k}_{\text{CAM}} := \frac{1}{W \times H} \sum_{w=1}^{W} \sum_{h=1}^{H} \frac{\partial y^c(x;\theta)}{\partial A^k_{w,h}} \; , \tag{18}$$

representing the overall importance of the $k$-th feature map for predicting class $c$.

The class-discriminative localization map $L^c_{\text{CAM}}(x)$ is then computed as a weighted combination of the activation maps, followed by upsampling to match the dimensions of the input image $x$:

$$L^c_{\text{CAM}}(x) := \mathcal{U}\left( \text{ReLU}\left( \sum_k \alpha^{c,k}_{\text{CAM}} A^k \right) \right) \; , \tag{19}$$

where $\mathcal{U}$ denotes the upsampling function.

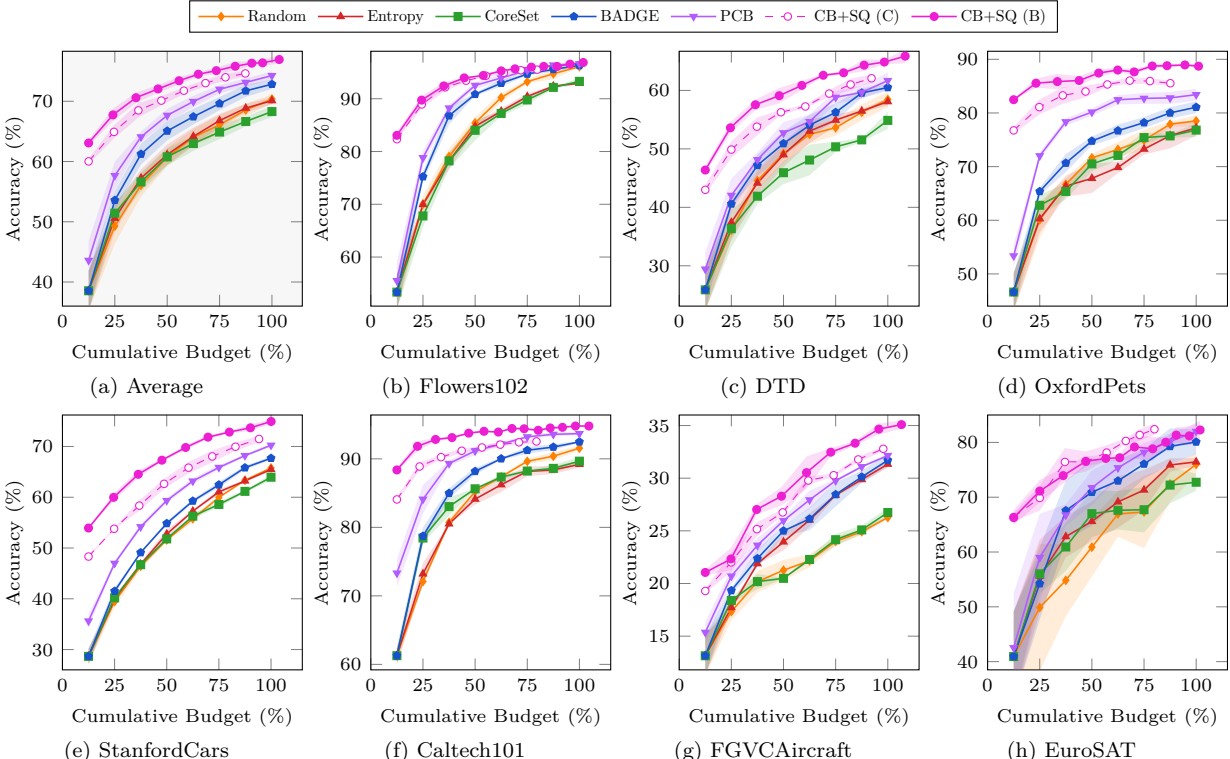

Figure 11: *Effect of the proposed method.* (a) Our CB+SQ outperforms the other baselines in average performance across 7 datasets.

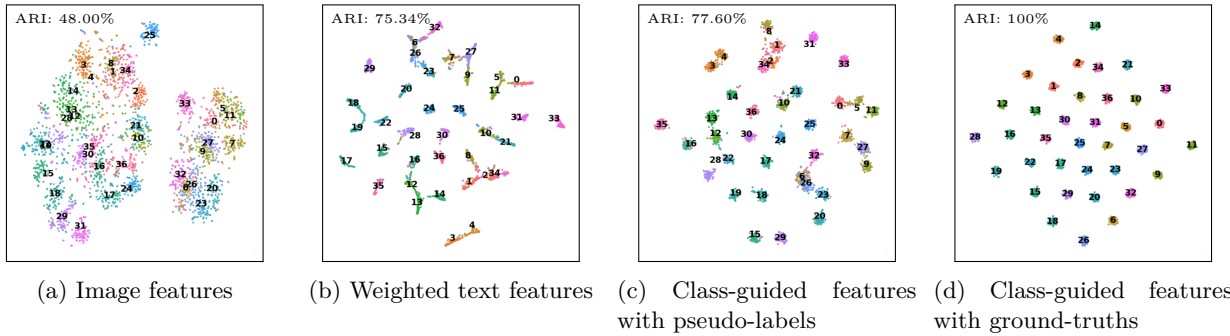

Figure 12: *Various clustering methods on OxfordPets dataset.* (a) Clustering based solely on image features primarily separates the data into two large groups, corresponding to dogs and cats, but fails to capture finer details. (b) Clustering on weighted text features improves Adjusted Rand Index (ARI), but some clusters remain ambiguous. (c) Class-guided clustering with pseudo-labels produces more distinct clusters, aligned with the guided class set $|\mathcal{C}| = 37$. (d) Class-guided clustering with ground truth shows perfect alignment of clusters.

**Gradient-weighted feature activation map (FAM).** To adapt GradCAM for vision-language models (VLMs), such as CLIP, which consist of an image encoder $\theta_{\text{img}}$ and a text encoder $\theta_{\text{txt}}$, we redefine the weight $\alpha$ with target features $\mathcal{F}_{\text{target}}$ as follows:

$$\alpha_{\text{FAM}}^k := \frac{1}{W \times H} \sum_{w=1}^{W} \sum_{h=1}^{H} \frac{\partial \cos(\theta_{\text{img}}(x), \mathcal{F}_{\text{target}})}{\partial A_{w,h}^k} , \tag{20}$$

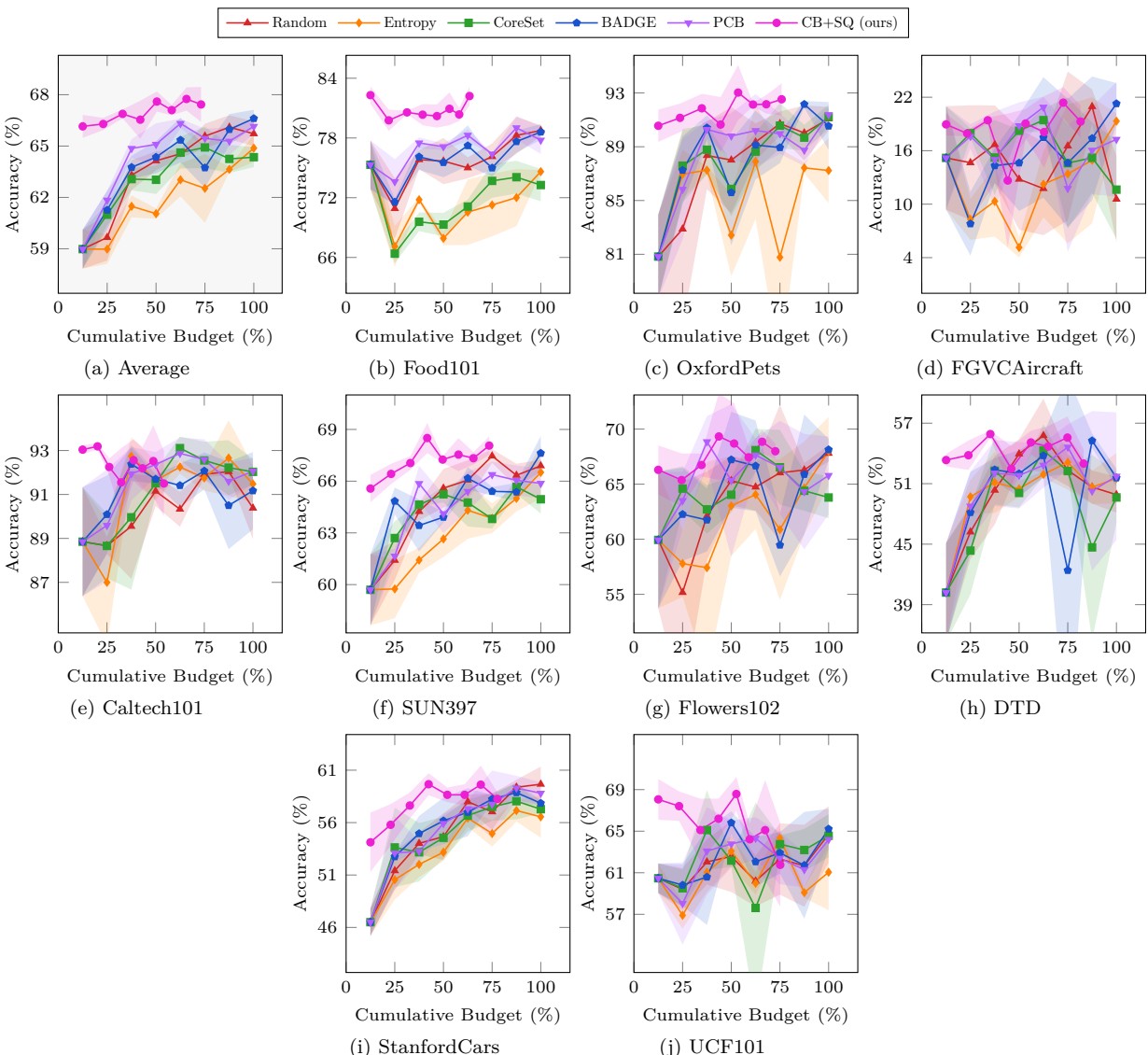

Figure 13: *Harmonic mean accuracy.* (a) Our CB+SQ outperforms the other baselines in average performance across 9 datasets. (b-j) The fluctuation in harmonic mean performance is attributed to variations in performance on novel classes, stemming from the limitations of the base prompt learning method, CoOp (Zhou et al., 2022b).

representing the overall important of the $k$-th feature map in determining the cosine similarity between the image features $\theta_{\text{img}}(x)$ and the target features $\mathcal{F}_{\text{target}}$. Here, the key difference from the GradCAM's weights in (18) lies in the absence of a specific class $c$. This modification enables the use of GradFAM for more flexible and label-independent analyses, accommodating the multimodal nature of VLMs. Building on this, the target feature discriminative localization map $L_{\text{FAM}}(x)$ is then computed as:

$$L_{\text{FAM}}(x) := \mathcal{U}\left(\text{ReLU}\left(\sum_k \alpha_{\text{FAM}}^k A^k\right)\right). \tag{21}$$

Our GradFAM can visualize the importance of various target features on the image, including (i) $\mathcal{F}_{\text{target}} = \theta_{\text{txt}}(c)$, (ii) $\mathcal{F}_{\text{target}} = \theta_{\text{img}}(x)$, and (iii) $\mathcal{F}_{\text{target}} = \mathcal{F}_{\mathcal{C}}(x)$. Specifically, when $\mathcal{F}_{\text{target}} = \theta_{\text{txt}}(c)$, our GradFAM operates almost identically to the original GradCAM. For $\mathcal{F}_{\text{target}} = \theta_{\text{img}}(x)$, due to VLMs being trained

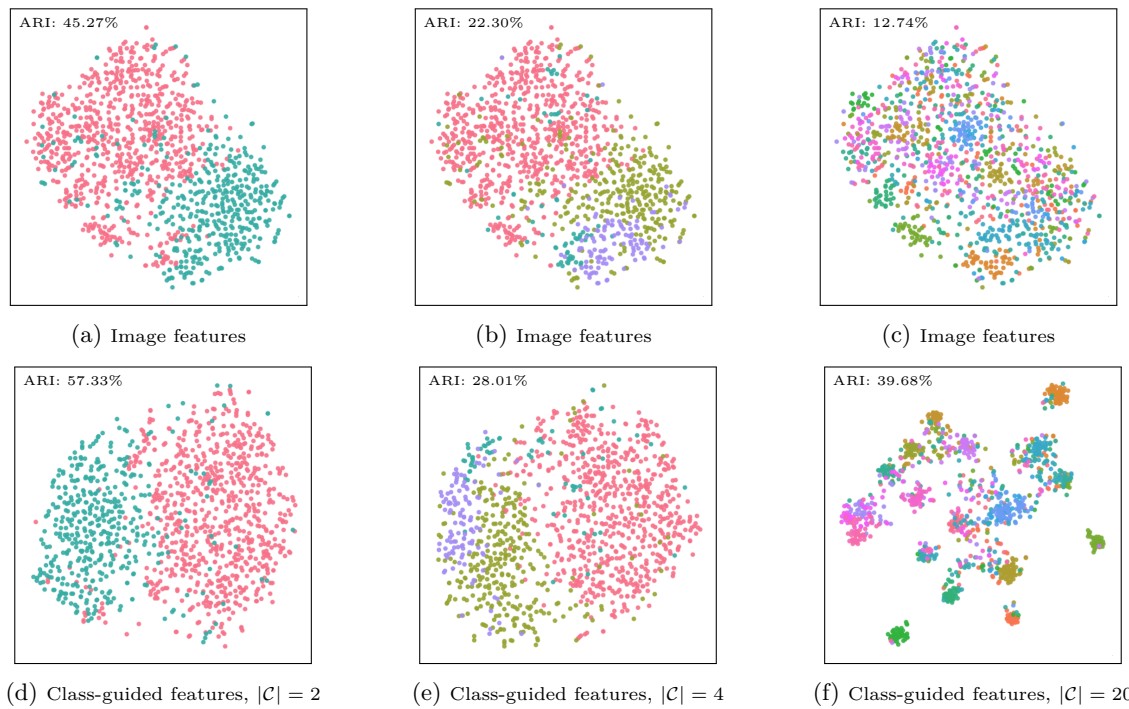

Figure 14: *Class-guided clustering on WaterBirds dataset.* (a, b, c) Clustering based solely on image features results in clusters that are poorly separated. (d, e, f) In contrast, our class-guided clustering, which incorporates class information, leads to more distinct clusters that align with the size of the guiding class set $\mathcal{C}$.

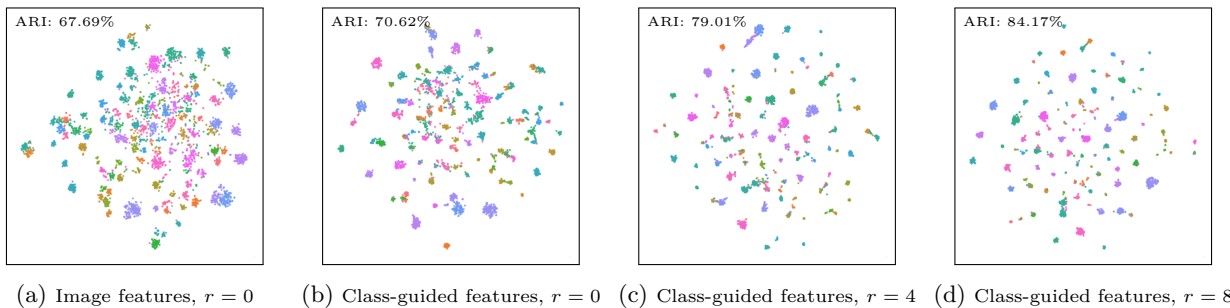

Figure 15: *Class-guided clustering across different rounds.* (a, b) In the initial round, *i.e.* $r = 0$, clustering on class-guided features achieves a higher ARI compared to clustering on image features. (c, d) We observe an improvement in ARI performance as the rounds progress.

through contrastive learning to align images and texts in a shared embedding space, all objects in the image are highlighted. In the case of our class-guided features $\mathcal{F}_{\mathcal{C}}(x)$ in (7), the guiding class set $\mathcal{C}$ effectively highlights the corresponding objects.

# B   Active Learning with Conserved Budgets

For fair comparisons, we evaluate our CB+SQ alongside with baselines over eight rounds, as illustrated in Figure 3. However, thanks to our selective querying in Section 4.3, we conserve budgets for each round. Here, we conduct additional experiments using the conserved budget. Figure 11 shows that our CB+SQ outperforms the baselines under various budget scenarios.

Table 4: *Examples of guiding class set.* To improve the clarity of clustering visualization, we randomly subsample 20 classes from the total of 200 classes.

| Index | Name | Index | Name | Index | Name | Index | Name |
|---|---|---|---|---|---|---|---|
| 019 | Gray Catbird | 027 | Shiny Cowbird | 029 | American Crow | 032 | Mangrove Cuckoo |
| 046 | Gadwall | 057 | Rose-breasted Grosbeak | 058 | Pigeon Guillemot | 061 | Heermann's Gull |
| 069 | Rufous Hummingbird | 073 | Blue Jay | 079 | Belted Kingfisher | 098 | Scott Oriole |
| 116 | Chipping Sparrow | 121 | Grasshopper Sparrow | 125 | Lincoln Sparrow | 155 | Warbling Vireo |
| 167 | Hooded Warbler | 172 | Nashville Warbler | 174 | Palm Warbler | 185 | Bohemian Waxwing |

Table 5: *Various guiding class sets.* Based on two habitats and two backgrounds, we construct class sets comprising 2 and 4 classes.

| $|\mathcal{C}|$ | $\mathcal{C}$ |
|---|---|
| 2 | land background |
|  | water background |
| 4 | land bird on land background |
|  | land bird on water background |
|  | water bird on land background |
|  | water bird on water background |

Table 6: *Various values of $K$ for $K$-means clustering.* Our progressively increasing $K$ across rounds achieves the best performance throughtout all rounds.

| Methods | Round 1 | Round 2 | Round 3 | Round 4 | Round 5 | Round 6 | Round 7 | Round 8 |
|---|---|---|---|---|---|---|---|---|
| $K = B$ | $63.65_{\pm0.16}$ | $66.37_{\pm0.16}$ | $67.07_{\pm0.49}$ | $68.34_{\pm0.46}$ | $69.15_{\pm0.16}$ | $69.43_{\pm0.36}$ | $69.96_{\pm0.10}$ | $70.28_{\pm0.21}$ |
| $K = 8 \times B$ | $59.77_{\pm0.27}$ | $64.30_{\pm0.59}$ | $67.33_{\pm0.57}$ | $69.30_{\pm0.48}$ | $70.62_{\pm0.63}$ | $71.92_{\pm0.54}$ | $72.46_{\pm0.32}$ | $72.96_{\pm0.26}$ |
| $K = r \times B$ | $\mathbf{63.85}_{\pm0.26}$ | $\mathbf{66.91}_{\pm0.38}$ | $\mathbf{68.92}_{\pm0.81}$ | $\mathbf{70.49}_{\pm0.06}$ | $\mathbf{71.21}_{\pm0.18}$ | $\mathbf{72.16}_{\pm0.28}$ | $\mathbf{72.94}_{\pm0.09}$ | $\mathbf{73.34}_{\pm0.12}$ |

## C   Analyses of Class-Guided Clustering

In Figure 4, we visualize the difference between class-guided clustering and conventional clustering based on image features. In this section, we present additional analyses on various clustering methods across different datasets and rounds.

**Class-guided clustering on OxfordPets dataset.** In Figure 12, we analyze various clustering methods, including clustering on (a) image features in (5), (b) weighted text features in (6), (c) class-guided features with pseudo-labels in (7), (d) class-guided features with ground-truth labels, where the weights of weighted text features are replaced to ground-truth labels, *i.e.* one-hot encodings on labels. Figure 12 illustrates that clustering on class-guided features achieves higher Adjusted Rand Index (ARI) values. Especially, Figure 12d suggests that as the performance of VLMs improves, perfect clustering becomes achievable.

**Class-guided clustering on WaterBirds dataset.** In Section 5.3 and Figure 4, we introduce the WaterBirds dataset to demonstrate the benefits of class-guided clustering, where different classes are guided within the same dataset. Specifically, the Waterbirds dataset comprises 200 distinct bird species, with each image annotated by habitat (water, land), background (water, land), and specific species. For our analyses, we select 20 classes and leverage various label information to separate the subset into groups of 2, 4, and 20 classes. Tables 4 and 5 provide the detailed class names. We note that text prompts such as "a photo of a ___ " are prepended to each class $c \in \mathcal{C}$ to generate final prompts. As shown in Figure 14, class-guided features based on different sizes of guiding class sets $\mathcal{C}$ effectively represent class-specific information.

**Cluster-Guided Clustering with Various Rounds.** We analyze the impact of class-guided clustering, derived from the zero-shot CLIP model, in comparison to various other clustering methods before initiating active learning. Here, we investigate cluster-guided clustering on the OxfordFlowers dataset, utilizing text

Table 7: *Experiments on the ISIC dataset.*

| Budget (%) | Entropy | CoreSet | BADGE | PCB | CB (ours) |
|---|---|---|---|---|---|
| 25 | 57.87 | 53.69 | 58.37 | 51.49 | **63.38** |
| 50 | 61.35 | 59.21 | 62.10 | 57.52 | **64.01** |
| 75 | 61.65 | 60.81 | 63.89 | 62.10 | **64.41** |
| 100 | 62.25 | 60.91 | 65.59 | 64.54 | **65.70** |

Table 8: *Experiments on the KaoKore dataset.*

| Budget (%) | Entropy | CoreSet | BADGE | PCB | CB (ours) |
|---|---|---|---|---|---|
| 25 | 51.68 | 45.75 | 45.36 | 50.74 | **52.62** |
| 50 | 56.36 | 53.32 | 55.58 | 56.52 | **56.83** |
| 75 | 57.73 | 57.06 | 59.02 | 57.30 | **59.33** |
| 100 | 60.11 | 59.25 | 61.01 | 60.58 | **61.90** |

Table 9: *Experiments on the DTD dataset with CLIP ViT-L/14-336.*

| Budget (%) | Entropy | CoreSet | BADGE | PCB | CB (ours) |
|---|---|---|---|---|---|
| 25 | 43.00 | 41.13 | 44.25 | 50.49 | **64.10** |
| 50 | 57.60 | 52.66 | 60.74 | 64.44 | **68.83** |
| 75 | 65.33 | 61.70 | 66.19 | 68.93 | **70.90** |
| 100 | 69.41 | 65.17 | 69.62 | 72.60 | **72.97** |

Table 10: *Experiments on the DTD dataset with EVA01-CLIP-g-14-plus (ViT-H/14).*

| Budget (%) | Entropy | CoreSet | BADGE | PCB | CB (ours) |
|---|---|---|---|---|---|
| 25 | 40.17 | 39.38 | 44.46 | 43.87 | **58.27** |
| 50 | 58.45 | 52.44 | 60.09 | 63.95 | **68.32** |
| 75 | 67.18 | 60.72 | 67.65 | 69.11 | **71.81** |
| 100 | 71.61 | 66.77 | 71.91 | 72.71 | **73.92** |

prompts that evolve with each round. Figure 15 illustrates that as the rounds progress, class-guided clustering forms increasingly well-separated clusters, accompanied by a steady increase in ARI.

**Effect of Various $K$.** In Section 4.2, we set $K$ equal to the budget $B$ in the initial round and introduce a linearly increasing $K$ according to round $r$, *i.e.* $K = r \times B$. Here, we analyze the effect of this increasing $K$. Table 6 shows that fixed values of $K$, whether small ($K = B$) or large ($K = 8 \times B$), are less effective compared to our incrementally increasing $K$. Specifically, using a small $K$ results in multiple samples being selected from the same cluster, leading to redundancy and reduced performance. On the other hand, a large $K$ fails to select representative samples during the initial round, resulting in diminished performance.

# D   Additional Ablation Studies

**Extension to Non-Natural Image Domains.** We conduct additional experiments on the medical dataset ISIC (Codella et al., 2019) in Table 7 and the illustrative dataset KaoKore (Tian et al., 2020) in Table 8. While the improvements are less pronounced than those on our main benchmarks in Figure 3, the proposed class-balanced (CB) acquisition still consistently outperforms strong baselines, indicating that the approach remains effective even when VLM priors are less directly aligned with the target domain.

**Generalization to Other Backbones.** We utilize CLIP ViT-B/32 architecture for main experiments. Here, we extend the evaluation to larger backbones: CLIP ViT-L/14-336 and EVA01-CLIP-g-14-plus (ViT-H/14) (Sun et al., 2023). Tables 9 and 10 show results on the DTD dataset. Here, we follows the same experimental setup with Figure 3, except for the backbone. Increasing model capacity generally improves

Table 11: *Effect of class-guided features on initial pool selection. In the initial round, incorporating class-guided features consistently improves baseline performance across six datasets. \* indicates that K-means clustering is employed.*

| Type | Method | Caltech101 | OxfordPets | StanfordCars | Flowers102 | FGVCAircraft | DTD | Avg |
|------|--------|-----------|------------|--------------|------------|--------------|-----|-----|
| – | Random | $62.10_{\pm 0.42}$ | $46.95_{\pm 0.88}$ | $30.56_{\pm 0.54}$ | $53.36_{\pm 3.01}$ | $14.64_{\pm 0.37}$ | $25.35_{\pm 1.98}$ | $38.83_{\pm 0.65}$ |
| Image Features | TypiClust | $77.05_{\pm 0.20}$ | $58.27_{\pm 0.17}$ | $36.53_{\pm 0.15}$ | $76.64_{\pm 0.16}$ | $17.46_{\pm 0.49}$ | $41.11_{\pm 0.28}$ | $51.18_{\pm 0.11}$ |
| Image Features | ProbCover | $51.68_{\pm 0.23}$ | $53.02_{\pm 1.04}$ | $28.71_{\pm 0.16}$ | $51.12_{\pm 0.38}$ | $16.49_{\pm 0.34}$ | $32.76_{\pm 1.04}$ | $38.96_{\pm 0.15}$ |
| Image Features | ProbCover* | $75.34_{\pm 1.01}$ | $55.73_{\pm 0.49}$ | $34.36_{\pm 0.07}$ | $72.24_{\pm 1.34}$ | $16.55_{\pm 0.75}$ | $38.77_{\pm 2.59}$ | $48.83_{\pm 0.19}$ |
| Image Features | DropQuery | $77.97_{\pm 0.37}$ | $57.43_{\pm 0.33}$ | $37.39_{\pm 0.14}$ | $75.30_{\pm 0.48}$ | $17.09_{\pm 0.10}$ | $42.10_{\pm 0.34}$ | $51.21_{\pm 0.01}$ |
| Class-guided Features | TypiClust | $81.08_{\pm 0.27}$ | $71.45_{\pm 1.57}$ | $40.87_{\pm 0.36}$ | $78.83_{\pm 0.14}$ | $18.10_{\pm 0.17}$ | $40.05_{\pm 0.74}$ | $\mathbf{55.06_{\pm 0.06}}$ |
| Class-guided Features | ProbCover | $63.95_{\pm 0.89}$ | $62.58_{\pm 1.91}$ | $32.61_{\pm 0.00}$ | $61.31_{\pm 0.32}$ | $15.81_{\pm 0.13}$ | $35.50_{\pm 0.51}$ | $\mathbf{46.99_{\pm 1.33}}$ |
| Class-guided Features | ProbCover* | $78.86_{\pm 1.06}$ | $68.61_{\pm 1.67}$ | $39.11_{\pm 0.17}$ | $77.07_{\pm 0.37}$ | $17.83_{\pm 0.62}$ | $38.40_{\pm 0.77}$ | $\mathbf{53.31_{\pm 0.28}}$ |
| Class-guided Features | DropQuery | $79.85_{\pm 0.32}$ | $70.95_{\pm 0.40}$ | $40.41_{\pm 0.53}$ | $77.86_{\pm 0.34}$ | $17.52_{\pm 0.36}$ | $39.11_{\pm 0.49}$ | $\mathbf{54.28_{\pm 0.10}}$ |

the final performance when the annotation budget is fully consumed, and CB remains consistently superior to baselines, supporting the generalizability of our acquisition strategy across VLM backbones.

**Additional Clustering-based Baselines with Class-guided Features.** In Table 11, we first evaluate these baselines in the initial round using only image features and find that they mitigate the cold-start issue more effectively than Random. When class-guided features are added, all baselines benefit from a positive synergy that further enhances performance. Specifically, TypiClust (Hacohen & Weinshall, 2023b), ProbCover (K-means) (Yehuda et al., 2022), and DropQuery (Gupte et al., 2024) employ K-means clustering in the given feature space, selecting samples within each cluster based on typicality, coverage, or proximity to the centroid, respectively. Our method operates in the same manner as DropQuery (Gupte et al., 2024) during initial pool selection, but differs in that we use class-guided features instead of image features. As shown in Table 11 and Figure 9c, the proposed class-guided features can be easily integrated into existing baselines and yield substantial performance gains.

