# OpenReview forum: "Active Prompt Learning with Vision-Language Model Priors"
_TMLR — Accepted by TMLR_

### Review · Reviewer_d98e · 2025-08-03

**Summary Of Contributions:**

The authors propose an active learning method that integrates class-guided clustering and selective querying to efficiently adapt CLIP with minimal annotation budget.
Experiments across seven datasets show that their approach achieves better performance and budget efficiency than existing baselines. The proposed data-centric approach is also shown to enhance state-of-the-art model-centric prompt learning methods, showing widespread potential.

**Audience:**

Yes

**Audience Explanation:**

1. The class-guided features seems to be effective to indicate target text regions.
2. The experiment results show the effectiveness of the proposed method, which can enhance 3 existing methods consistently among 7 benchmarks.
3. The proposed method shows good generalization to unseen classes in the first round.

**Claims And Evidence:**

No

**Claims Explanation:**

I mainly have a concern for the second contribution of GradFAM.
While GradFAM provides a useful visualization of the proposed class-guided features, it appears to be primarily an analytical tool rather than a core methodological contribution. Since it is listed as one of the main contributions of the paper, I would appreciate clarification on the following points:
 - What specific limitations of existing feature analysis methods (e.g., GradCAM or other CLIP-based visual attribution) motivated the design of GradFAM? Could existing methods not be adapted for this purpose?
 - Are there prior works or techniques (e.g., CLIP-GradCAM, Attention Rollout) that have been proposed for feature-level analysis in VLMs? If so, how does GradFAM compare?
 - Does GradFAM provide any analytical or quantitative insight beyond qualitative visualization?

**Requested Changes:**

1. If the author wants to keep the second contribution, please add more supportive evidence, according to my comments above.
2. In GradFAM, how is the similarity $\cos(\mathcal{I}(x), \mathcal{F}_{\text{target}})$ computed when the target feature  is the class-guided feature, which is a concatenation of image and text features, resulting in a dimension twice that of $\mathcal{I}(x)$, making the cosine similarity unclear.

---

> ### Author Response · Authors · 2025-08-11
> **Response to Reviewer d98e (Part 1/2)**
>
> ### **General Reply**
> We appreciate the reviewer’s thoughtful and constructive feedback, which will help us improve our paper. Below, we respond to each of your comments in turn. We are happy to provide further clarification if needed.
>
> ### **Weakness 1. Clarification on Our Second Contribution**
>
> > I mainly have a concern for the second contribution of GradFAM. While GradFAM provides a useful visualization of the proposed class-guided features, it appears to be primarily an analytical tool rather than a core methodological contribution.
>
> > If the author wants to keep the second contribution, please add more supportive evidence, according to my comments above.
>
> Before addressing each comment individually, we would like to clarify our second contribution. Our second contribution focuses less on GradFAM itself and more on the in-depth qualitative analyses using GradFAM and t-SNE, as illustrated in Figures 2 and 4. Specifically, Figure 2 illustrates which parts of an image should be attended to during clustering. When clustering is conducted without a specific condition, as in Figure 2(b), the clustering focus on all objects within the image. In contrast, when there is a defined clustering objective, such as grouping by colors, it becomes necessary to focus only on the relevant regions, as shown in Figure 2(e). Moreover, Figure 4 presents the actual clustering results using t-SNE visualizations and ARI scores, demonstrating how the outcomes vary depending on the clustering objective and the number of clusters, such as 2 or 20. These results highlight the advantages of class-guided features in producing more meaningful clustering outcomes.
>
> ### **Weakness 2. Limitation of Existing Feature Analysis Methods**
>
> > What specific limitations of existing feature analysis methods (e.g., GradCAM or other CLIP-based visual attribution) motivated the design of GradFAM? Could existing methods not be adapted for this purpose?
>
> The major limitation of GradCAM [1] and CLIP-GradCAM [2] is that they require a single target class or text. These methods primarily focus on analyzing the visual contribution of a specific class or text, which becomes particularly challenging when applied to our active learning scenarios that start with unlabeled images. To address this, we extend the GradCAM into GradFAM, focusing on target features, that operates even when 1) only the image is provided or 2) only the image and a class-set are available. Even in these cases, GradFAM enables analyses like those shown in (1) Figure 2(b) and (2) Figure 2(c-e), respectively.
>
> **References**
> * [1] Grad-CAM: Visual Explanations from Deep Networks via Gradient-based Localization, IJCV 2019.
> * [2] https://colab.research.google.com/github/kevinzakka/clip_playground/blob/main/CLIP_GradCAM_Visualization.ipynb
>
> ### **Weakness 3. Comparisons with Prior Works**
>
> > Are there prior works or techniques (e.g., CLIP-GradCAM, Attention Rollout) that have been proposed for feature-level analysis in VLMs? If so, how does GradFAM compare?
>
> A straightforward baseline is to derive a pseudo-label from the given class set and apply CLIP-GradCAM [1]. However, in the second row of Figure 2, when the class set is {dog, pot}, the pseudo-label for the original image is dog, so CLIP-GradCAM highlights only the dog region. In contrast, as we use the weighted text feature, our method can highlight both the dog and the pot.
>
> Next, Attention Rollout [2] explains the model's internal information flow by visualizing how attention is propagated within the Transformer architecture to reveal its baseline behavior. In contrast, GradFAM is specifically designed to visualize the spatial influence of externally defined features, such as our proposed class-guided features. Thus, the two methods are complementary: while Attention Rollout can offer a general view of the model's inherent focus, GradFAM provides a targeted analysis of how our specific methodological contribution guides the model's attention, enabling it to highlight multiple concepts based on a given class set.
>
> **References**
> * [1] https://colab.research.google.com/github/kevinzakka/clip_playground/blob/main/CLIP_GradCAM_Visualization.ipynb
> * [2] Quantifying Attention Flow in Transformers, ACL 2020.

---

> > ### Author Response · Authors · 2025-08-11
> > **Response to Reviewer d98e (Part 2/2)**
> >
> > ### **Weakness 4. Beyond Qualitative Visualization**
> >
> > > Does GradFAM provide any analytical or quantitative insight beyond qualitative visualization?
> >
> > We currently provide only qualitative visualizations with GradFAM. However, the results in Figure 2(b) suggest that the method could be extended to yield analytical or quantitative insights in the future. In Figure 2(b), the results are obtained using only the image encoder of a pretrained CLIP model, and the outcomes are expected to vary with the choice of encoder. For example, an image encoder optimized for animal classification would likely highlight primarily the animal regions, as its image features vary more when the animal object changes than when other objects including the background change. This indicates a path toward label-free analysis of encoder bias by checking which regions the encoder emphasizes. However, we note that this is challenging to realize immediately and regard it as a promising avenue for future application.
> >
> > ### **Weakness 5. Clarifying Implementation Details for Visualizations**
> >
> > > In GradFAM, how is the similarity  computed when the target feature is the class-guided feature, which is a concatenation of image and text features, resulting in a dimension twice that of , making the cosine similarity unclear.
> >
> > Thank you for flagging the potentially confusing point. We concatenate two 512-dimensional image features to form a 1024-dimensional representation, and use the class-guided feature as the target feature.

---

### Review · Reviewer_WqbA · 2025-08-03

**Summary Of Contributions:**

The paper proposes a data‑centric active‑learning framework that adapts VLMs to downstream datasets with minimal annotation cost. The work uses class-guided feature construction followed by a linearly growing K-means clustering to obtain a cluster‑balanced candidate pool. It also utilizes pseudo-labels to assist in the selective querying process for active learning. They evaluate their proposed approach across seven image‑classification benchmarks, showing higher accuracy than Random, Entropy, CoreSet, BADGE and PCB while reducing human labeling.

Strength:

1. The cold-start problem is well handled by utilizing VLM predictions as priors.
2. The modified GradFAM for annotation‑free saliency is done well. ARI and t‑SNE analyses show class‑guided clustering is semantically aligned.
3. The proposed active prompt learning reduces the need for human querying by ~18%.
4. The evaluation in Figure 3 shows boost in performance across almost all the datasets.

Weakness:

1. The work relies much on the zero-shot priors which doesnot always adapt well to downstream datasets, especially not having natural images. The evaluation of the proposed approach on other types datasets such as medical datasets (ISIC [1]),  illustrative datasets (Kaokore [2]]) and structured datasets (Clevr-Count [1]) is crucial to gauge the contribution of the proposed prompt active learning approach.

2. In this view, the proposed approach also shows reliance on only CLIP model as a supporting VLM. The authors should also highlight the performance using other backbones such as EVA‑CLIP. Does the approach only function well using CLIP? These questions the generalizability of the approach.

3. More details on the robustness analysis needs to be provided regarding the performance tied to linear K‑schedule, prompt re‑initialisation and threshold formula. How does the pseudo-label threshold affect the performance? Does one threshold perform well across all the datasets? Including an ablation study for the changing pseudo-label threshold can give a better clarity.

4. In regard of the ImageNet experiments, the compute overhead is opaque. What is the compute requirement and GPU hours required for K‑means on millions of ImageNet embeddings plus per‑round prompt retraining which is certainly not trivial.

5. The Base-to-novel generalization including Table 3 seems confusing. Where are the results for individual datasets? Does table 3 show only the average of all datasets?

[1] Codella, Noel, et al. "Skin lesion analysis toward melanoma detection 2018: A challenge hosted by the international skin imaging collaboration (isic)." arXiv preprint arXiv:1902.03368 (2019).

[2] Tian, Yingtao, et al. "Kaokore: A pre-modern japanese art facial expression dataset." arXiv preprint arXiv:2002.08595 (2020).

[3] Johnson, Justin, et al. "Clevr: A diagnostic dataset for compositional language and elementary visual reasoning." Proceedings of the IEEE conference on computer vision and pattern recognition. 2017.

**Audience:**

Yes

**Audience Explanation:**

The approach couples the essence of prompt learning with active learning. It would be interesting to venture more in this frontier.

**Broader Impact Concerns:**

Nil

**Claims And Evidence:**

Yes

**Claims Explanation:**

The claims are backed by suitable experiments. However, the key contributions of each module, including the GradFAM, need to be portrayed more clearly, including more ablation studies.

**Requested Changes:**

Please go through the Weakness section to find the detail suggestions and changes. In brief, the suggested changes are as follows:

1. Evaluate the performance on various datasets and using different model backbone.
2. Threshold Robustness – What happens if initial labeled data are noisy or class confidence is skewed (long‑tail classes)?
3. Hyperparameter Search Cost—K‑schedule, budget B, prompt length—how sensitive are final results, and are they tuned per dataset?

---

> ### Author Response · Authors · 2025-08-11
> **Response to Reviewer WqbA (Part 1/4)**
>
> ### **General Reply**
> We greatly appreciate all the valuable feedback and constructive suggestions, which we will incorporate to strengthen our paper. Each comment is addressed in detail below. Please feel free to request additional clarifications.
>
> ### **Weakness 1. Generalization to Non-Natural Image Domains**
>
> > The work relies much on the zero-shot priors which doesnot always adapt well to downstream datasets, especially not having natural images. The evaluation of the proposed approach on other types datasets such as medical datasets (ISIC [1]), illustrative datasets (Kaokore [2]]) and structured datasets (Clevr-Count [1]) is crucial to gauge the contribution of the proposed prompt active learning approach.
>
> > Evaluate the performance on various datasets and using different model backbone.
>
> Since we actively leverage VLM priors, performance improvements may be limited on downstream datasets where effectively utilizing these priors is challenging. Following your suggestion, we conduct additional experiments on the medical dataset ISIC [1] and the illustrative dataset KaoKore [2]. The results are shown in Table A and Table B, respectively. Although the performance gains are not as substantial as those observed on the original benchmarks in Figure 3, these results still demonstrate the effectiveness of our approach.
>
> **Table A.** *Experimental results on the ISIC dataset.* We compare our class-balanced (CB) acquisition against other baselines, excluding our selective querying (SQ) technique. The experimental setup follows the same conditions as those in Figure 3.
> | Budget (%) | Entropy | Coreset | BADGE | PCB   | CB (ours) |
> |:----------:|:-------:|:-------:|:-----:|:-----:|:----------:|
> | 25         | 57.87   | 53.69   | 58.37 | 51.49 | **63.38**  |
> | 50         | 61.35   | 59.21   | 62.10 | 57.52 | **64.01**  |
> | 75         | 61.65   | 60.81   | 63.89 | 62.10 | **64.41**  |
> | 100       | 62.25   | 60.91   | 65.59 | 64.54 | **65.70**  |
>
> **Table B.** *Experimental results on the KaoKore dataset.* We compare our class-balanced (CB) acquisition against other baselines, excluding our selective querying (SQ) technique. The experimental setup follows the same conditions as those in Figure 3.
> | Budget (%) | Entropy | Coreset | BADGE | PCB   | CB (ours) |
> |:----------:|:-------:|:-------:|:-----:|:-----:|:----------:|
> | 25         | 51.68   | 45.75   | 45.36 | 50.74 | **52.62**  |
> | 50         | 56.36   | 53.32   | 55.58 | 56.52 | **56.83**  |
> | 75         | 57.73   | 57.06   | 59.02 | 57.30 | **59.33**  |
> | 100       | 60.11   | 59.25   | 61.01 | 60.58 | **61.90**  |
>
> **References**
> * [1] Skin Leison Analysis Toward Melanoma Detection 2018: A Challenge Hosted by the International Skin Imaging Collaboration (ISIC), arXiv 2019.
> * [2] KaoKore: A Pre-modern Japanese Art Facial Expression Dataset, ICCC 2020.

---

> > ### Author Response · Authors · 2025-08-11
> > **Response to Reviewer WqbA (Part 2/4)**
> >
> > ### **Weakness 2. Generalization to Other Backbones**
> >
> > > In this view, the proposed approach also shows reliance on only CLIP model as a supporting VLM. The authors should also highlight the performance using other backbones such as EVA‑CLIP. Does the approach only function well using CLIP? These questions the generalizability of the approach.
> >
> > > Evaluate the performance on various datasets and using different model backbone.
> >
> > Our initial experiments were conducted using only the CLIP ViT-B/32 model. To address this, we extend our evaluation to include CLIP with a ViT-L/14-336 backbone and EVA01-CLIP-g-14-plus with a ViT-H/14 backbone [1]. The corresponding results are presented in Table C and Table D, respectively. Increasing the model size generally improved the final performance when the budget are fully consumed. The results demonstrate the generalizability of our approach across multiple backbone architectures.
> >
> > **Table C.** *Experimental results on the DTD dataset using CLIP with a ViT-L/14-336.* We compare our class-balanced (CB) acquisition against other baselines, excluding our selective querying (SQ) technique. The experimental setup follows the same conditions as those in Figure 3-(c), except for the backbone architecture.
> > | Budget (%) | Entropy | Coreset | BADGE | PCB   | CB (ours) |
> > |:----------:|:-------:|:-------:|:-----:|:-----:|:----------:|
> > | 25         | 43.00 | 41.13 | 44.25 | 50.49 | **64.10** |
> > | 50         | 57.60 | 52.66 | 60.74 | 64.44 | **68.83** |
> > | 75         | 65.33 | 61.70 | 66.19 | 68.93 | **70.90** |
> > | 100       | 69.41 | 65.17 | 69.62 | 72.60 | **72.97** |
> >
> > **Table D.** *Experimental results on the DTD dataset using EVA01-CLIP-g-14-plus with a ViT-H/14.* We compare our class-balanced (CB) acquisition against other baselines, excluding our selective querying (SQ) technique. The experimental setup follows the same conditions as those in Figure 3-(c), except for the backbone architecture.
> > | Budget (%) | Entropy | Coreset | BADGE | PCB   | CB (ours) |
> > |:----------:|:-------:|:-------:|:-----:|:-----:|:----------:|
> > | 25         | 40.17 | 39.38 | 44.46 | 43.87 | **58.27** |
> > | 50         | 58.45 | 52.44 | 60.09 | 63.95 | **68.32** |
> > | 75         | 67.18 | 60.72 | 67.65 | 69.11 | **71.81** |
> > | 100       | 71.61 | 66.77 | 71.91 | 72.71 | **73.92** |
> >
> > **References**
> > * [1] EVA-CLIP: Improved Training Techniques for CLIP at Scale, arXiv 2023.

---

> > > ### Author Response · Authors · 2025-08-11
> > > **Response to Reviewer WqbA (Part 3/4)**
> > >
> > > ### **Weakness 3. Additional Robustness Analyses**
> > >
> > > > More details on the robustness analysis needs to be provided regarding the performance tied to linear K‑schedule, prompt re‑initialisation and threshold formula.
> > >
> > > > Hyperparameter Search Cost—K‑schedule, budget B, prompt length—how sensitive are final results, and are they tuned per dataset?
> > >
> > > Previous study reports that skipping re-initialization across active learning rounds reduces training time but degrades performance [1]. Accordingly, we re-initialize the prompt at every round in our experiments. In Table 6 on page 21, $K$-schedule ablations show that smaller $K$ is preferable early under tight budgets, whereas larger $K$ becomes advantageous as the total budget grows. In addition, Figure 3 demonstrates the advantages of our method across budgets, and Table E further demonstrate robustness to the choice of prompt length. For most experiments in the paper, we set $K = r \times B$ with $r$ round index and $B = |\mathcal{C}|$ and fix the prompt length to 16, and apply no dataset-specific hyperparameter tuning.
> > >
> > > **Table E.** *Effect of Prompt Length on the DTD dataset.*  We compare our class-balanced (CB) acquisition against other baselines, excluding our selective querying (SQ) technique. The experimental setup follows the same conditions as those in Table C, except for the prompt length.
> > > | Prompt Length (%) | Entropy | Coreset | BADGE | PCB   | CB (ours) |
> > > |:----------:|:-------:|:-------:|:-----:|:-----:|:----------:|
> > > | 4         | 67.93 | 66.88 | 69.66 | 72.22 | **72.75** |
> > > | 8         | 67.79 | 66.21 | 67.85 | 72.30 | **73.54** |
> > > | 16       | 69.41 | 65.17 | 69.62 | 72.60 | **72.97** |
> > >
> > > > How does the pseudo-label threshold affect the performance? Does one threshold perform well across all the datasets? Including an ablation study for the changing pseudo-label threshold can give a better clarity.
> > >
> > > In the case of pseudo-label threshold, setting it too high diminishes budget savings, whereas setting it too low introduces noisy labels and degrades performance, as shown in Table F. In addition, as VLM priors vary widely across downstream tasks, a single fixed threshold is brittle. To address this, we use the threshold in Equation (11), which is hyperparameter-free and adapts automatically not only to each dataset but also to each class.
> > >
> > > **Table F.** *Effect of Pseudo-label Thresholds on the DTD dataset.* We first train a model on 4-shot data, use it to select the next 4-shot batch, and apply pseudo-labels only to samples whose scores exceed class-wise thresholds computed from Equation (11). Using these thresholds (mean 0.97; min 0.90; max 0.99), we achieve higher pseudo-label accuracy than fixed 0.90 or 0.95 thresholds, while requiring less budget than a fixed 0.99 threshold.
> > > | Threshold | Pseudo Label Accuracy (%) | Budget (%) | Performance (%) |
> > > |:---------:|:-------------------------:|:----------:|:----------------:|
> > > | 0.90 | 0.88 | 93.1 | 64.83 |
> > > | 0.95 | 0.95 | 94.9 | 65.01 |
> > > | 0.99 | 1.0  | 97.6 | 65.60 |
> > > | Ours | 1.0  | 95.4 | 65.60 |
> > >
> > > > Threshold Robustness – What happens if initial labeled data are noisy or class confidence is skewed (long‑tail classes)?
> > >
> > > Vision–language models like CLIP are known to exhibit class-dependent confidence [2]. To address this, we introduce Equation (11), a hyperparameter-free procedure that produces class-wise thresholds. On the DTD dataset, we observe that the obtained threshold lies between 0.72 and 0.95 across classes. As a result, we suppress pseudo-labels for easy classes even when confidence exceeds 0.9, while assigning pseudo-labels to hard classes at confidences as low as 0.8.
> > >
> > > **References**
> > > * [1] On Warm-Starting Neural Network Training, NeurIPS 2020.
> > > * [2] Active Prompt Learning in Vision Language Models, CVPR 2024.

---

> > > > ### Author Response · Authors · 2025-08-11
> > > > **Response to Reviewer WqbA (Part 4/4)**
> > > >
> > > > ### **Weakness 4. Computational Details for ImageNet Experiments**
> > > >
> > > > > In regard of the ImageNet experiments, the compute overhead is opaque. What is the compute requirement and GPU hours required for K‑means on millions of ImageNet embeddings plus per‑round prompt retraining which is certainly not trivial.
> > > >
> > > > The experimental pipeline on ImageNet consists of three primary steps. First, 1024-dimensional class-guided features are extracted from all ImageNet images, which takes approximately 4 hours. Second, we perform K-means clustering on these features with K=1000 in the initial round, requiring roughly 2 hours. Third, we conduct prompt learning using labeled 1-shot images, which takes about 3 hours. However, as active learning progresses, the value of K increases in subsequent rounds, resulting in a linear increase in the runtime of the K-means algorithm and prompt learning. All experiments are performed on a single RTX A6000 GPU and an AMD EPYC 7513 32-Core Processor.
> > > >
> > > > ### **Weakness 5. Base-to-novel Generalization Experiments**
> > > >
> > > > > The Base-to-novel generalization including Table 3 seems confusing. Where are the results for individual datasets? Does table 3 show only the average of all datasets
> > > >
> > > > The results presented in Table 3 represent the average performance across the nine datasets. Figure 13 on page 19 of the updated PDF illustrates the round-by-round harmonic mean performance for each individual dataset. It is important to note that the harmonic mean measures the combined performance of both base and novel classes. However, since active learning is conducted exclusively on the base classes, performance on the novel classes fluctuates significantly. As a result, the harmonic mean exhibits considerable variation across different rounds. Nonetheless, based on the average results shown in Figure 13(a), we emphasize that our CB+SQ approach consistently outperforms the other baseline methods.

---

### Review · Reviewer_2j62 · 2025-08-03

**Summary Of Contributions:**

This paper aims to address the data selection in active learning for few-shot prompt tuning for vision language models based on pre-trained prior knowledge. Specifically, it fuses the image-text features using weighted class-guided clustering and querying the samples to annotate based on a class adaptive threshold. The experimental results on standard benchmark demonstrates the effectiveness of the proposed method given the budget constraint.

The motivation is clear. Frames few-shot prompt tuning as active data selection, tackling cold-start with class-guided clustering.

Performance: seven datasets plus ImageNet show accuracy gains and label savings over baselines. Cluster sampling plus selective querying saves $\sim 17\%$ annotation cost by using reliable pseudo-labels.

Analysis shows GradFAM and t-SNE confirm the method creates better class separation.

**Audience:**

Yes

**Audience Explanation:**

The few-shot prompt tuning task is a little outdated, as the current state-of-the-art VLLMs can perform well as long as the pretraining data has sufficient coverage. There are already some papers exploring active learning for prompt tuning, and, furthermore, the findings from the traditional active learning field can be applied to the vision-language setting with minor modifications.

**Broader Impact Concerns:**

Considering all the above factors, I consider that the impact of this work might be a little limited.

**Claims And Evidence:**

Yes

**Claims Explanation:**

The active learning strategy is too simple. See requested changes for more details.

**Requested Changes:**

**The task setting scope is too narrow in practice.** Active learning for few-shot prompt tuning is demonstrated primarily on recognition tasks, but whether this method can generalize to broader task categories beyond recognition remains unclear.

**Incomplete evaluation on standard benchmarks.** Why are the complete comparison results on the ImageNet dataset not presented? This omission makes it difficult to assess the method's performance against established baselines on a widely-used benchmark.

**Unexplained performance characteristics.** The proposed method appears to saturate faster than other methods in subfigures (b, d, f, h) of Figure 3. What causes this early saturation, and does this indicate a fundamental limitation of the approach?

**Limited technical novelty.** The proposed class-guided clustering technique shares similar principles with existing semantic feature fusion and augmentation techniques, such as CLIP-GCD [1]. Additionally, while different active learning papers typically propose distinct sampling and querying strategies, the authors do not conduct comprehensive ablations comparing different active labeling strategies, which limits our understanding of the method's core contributions.

**Reference:**
[1] CLIP-GCD: Simple Language Guided Generalized Category Discovery

---

> ### Author Response · Authors · 2025-08-11
> **Response to Reviewer 2j62 (Part 1/2)**
>
> ### **General Reply**
> We sincerely appreciate all the insightful and constructive comments, which we will incorporate to further improve our paper. Below, we address each comment individually. Please let us know if any further clarification is required.
>
> ### **Weakness 1. Broad Applicability in Recognition Tasks**
>
> > The task setting scope is too narrow in practice. Active learning for few-shot prompt tuning is demonstrated primarily on recognition tasks, but whether this method can generalize to broader task categories beyond recognition remains unclear.
>
> Our method is challenging to extend beyond image recognition tasks, however, we believe that recognition tasks are an important and broad area. To cover this broad area more comprehensively, we also conduct experiments on additional datasets representing different visual recognition scenarios. Specifically, we evaluate our method on the ISIC dataset [1], a medical image dataset consisting of dermatological images of skin lesions for melanoma classification, and the KaoKore dataset [2], an illustrative dataset containing face images from traditional Japanese artworks. The results are shown in Table A and Table B, respectively. Although the performance gains are not as substantial as those observed on the original benchmarks in Figure 3, these results demonstrate the board applicability of our method.
>
> **Table A.** *Experimental results on the ISIC dataset.* We compare our class-balanced (CB) acquisition against other baselines, excluding our selective querying (SQ) technique. The experimental setup follows the same conditions as those in Figure 3.
> | Budget (%) | Entropy | Coreset | BADGE | PCB   | CB (ours) |
> |:----------:|:-------:|:-------:|:-----:|:-----:|:----------:|
> | 25         | 57.87   | 53.69   | 58.37 | 51.49 | **63.38**  |
> | 50         | 61.35   | 59.21   | 62.10 | 57.52 | **64.01**  |
> | 75         | 61.65   | 60.81   | 63.89 | 62.10 | **64.41**  |
> | 100       | 62.25   | 60.91   | 65.59 | 64.54 | **65.70**  |
>
> **Table B.** *Experimental results on the KaoKore dataset.* We compare our class-balanced (CB) acquisition against other baselines, excluding our selective querying (SQ) technique. The experimental setup follows the same conditions as those in Figure 3.
> | Budget (%) | Entropy | Coreset | BADGE | PCB   | CB (ours) |
> |:----------:|:-------:|:-------:|:-----:|:-----:|:----------:|
> | 25         | 51.68   | 45.75   | 45.36 | 50.74 | **52.62**  |
> | 50         | 56.36   | 53.32   | 55.58 | 56.52 | **56.83**  |
> | 75         | 57.73   | 57.06   | 59.02 | 57.30 | **59.33**  |
> | 100       | 60.11   | 59.25   | 61.01 | 60.58 | **61.90**  |
>
> **References**
> * [1] Skin Leison Analysis Toward Melanoma Detection 2018: A Challenge Hosted by the International Skin Imaging Collaboration (ISIC), arXiv 2019.
> * [2] KaoKore: A Pre-modern Japanese Art Facial Expression Dataset, ICCC 2020.

---

> > ### Author Response · Authors · 2025-08-11
> > **Response to Reviewer 2j62 (Part 2/2)**
> >
> > ### **Weakness 2. Limited ImageNet Evaluation due to Scale**
> >
> > > Incomplete evaluation on standard benchmarks. Why are the complete comparison results on the ImageNet dataset not presented? This omission makes it difficult to assess the method's performance against established baselines on a widely-used benchmark.
> >
> > Although ImageNet is widely-used as a standard benchmark dataset, it is generally too large for active learning experiments. Therefore, even baseline methods such as Coreset [1] and BADGE [2] do not conduct experiments at ImageNet scale. Previous work [3] also indicates that applying Coreset and BADGE at ImageNet scale is infeasible, and the BADGE-based PCB [4] faces the same limitation. However, experiments with computationally lighter methods, such as Random and Entropy, as well as our proposed approach CB+SQ, are feasible at ImageNet scale. Accordingly, Figure 8 shows that CB+SQ outperforms these two baselines.
> >
> > **References**
> > * [1] Active Learning for Convolutional Neural Networks: A Core-Set Approach, ICLR 2018.
> > * [2] Deep Batch Active Learning by Diverse, Uncertain Gradient Lower Bounds, ICLR 2020.
> > * [3] Active Learning at the ImageNet Scale, arXiv 2021.
> > * [4] Active Prompt Learning in Vision Language Models, CVPR 2024.
> >
> > ### **Weakness 3. Perceived Saturation due to Early Oracle Convergence**
> >
> > > Unexplained performance characteristics. The proposed method appears to saturate faster than other methods in subfigures (b, d, f, h) of Figure 3. What causes this early saturation, and does this indicate a fundamental limitation of the approach?
> >
> > Some subfigures in Figure 3 appear to saturate faster, but this phenomenon is related to the oracle accuracy, which represents the maximum achievable accuracy obtained by labeling all images without any budget constraints. For example, the oracle performance is 97.97% for (b) Flowers102, 90.16% for (d) OxfordPets, 95.21% for (f) Caltech101, and 94.83% for (h) EuroSAT. In these cases, our CB+SQ approach quickly achieves performance close to the oracle under limited budget conditions, giving the impression of saturation. For EuroSAT, the gap seems larger; however, this is because the oracle accuracy for EuroSAT is derived using 100 times more labeled images compared to our setting. Additionally, as illustrated in Figure 11 on page 18, the accuracy of CB+SQ continues to gradually improve even beyond the saturation point.
> >
> > ### **Weakness 4. Clarification on Technical Novelty**
> >
> > > Limited technical novelty. The proposed class-guided clustering technique shares similar principles with existing semantic feature fusion and augmentation techniques, such as CLIP-GCD [1]. Additionally, while different active learning papers typically propose distinct sampling and querying strategies, the authors do not conduct comprehensive ablations comparing different active labeling strategies, which limits our understanding of the method's core contributions.
> >
> > Thank you for recommending a new paper for the related work section. We will reflect this suggestion in our revision. Our approach differs conceptually from CLIP-GCD [1] by explicitly addressing the issue of selecting which images should be prioritized for annotation. Technically, rather than solely relying on text features, our method employs weighted text features that consider similarity to image features, as illustrated in Figure 1(a). Furthermore, we propose cluster-based sampling in Figure 1(b) and selective querying in Figure 1(c), tailored specifically for the active learning scenario, to improve the active labeling process. Specifically, we progressively increase the number of clusters in each round to ensure that some clusters remain unselected within the allocated budget. Moreover, unlike prior threshold-based methods that rely on a fixed hyperparameter applied uniformly across datasets, our approach is hyperparameter-free and employs an adaptive thresholding mechanism tailored to each class within the dataset, as shown in Equation (11).
> >
> > In Figure 3, we compare our method with active learning baselines from a sampling perspective, however as you pointed out, we do not provide comparisons from a querying standpoint. This is primarily due to the fact that these baseline methods do not propose additional querying strategies. To further clarify the contribution of querying, we demonstrate the synergy between our selective querying method and existing sampling baselines in Figures 5(a) and (b), particularly highlighting strong compatibility with diversity-based sampling methods. Additionally, in Figure 9(b), we analyze the individual contributions of our cluster-based sampling and selective querying components.
> >
> > **References**
> > * [1] CLIP-GCD: Simple Language Guided Generalized Category Discovery, arXiv 2023.

---

### Review · Reviewer_9bb1 · 2025-08-07

**Summary Of Contributions:**

- This paper introduces a low budget active learning method for image classification tasks with vision language models via prompt tuning. The acquisition function is based on cluster balanced sampling, and selective querying to avoid wasting the annotation budget on less uncertain samples.
- An important component of this method is cluster balanced sampling which also addresses the cold start problem in active learning. Instead of using just the features from the VLM's image encoder, the authors propose a weighting scheme to incorporate text features as well and combine both to generate class guided features which can then be clustered for balanced sampling.
- Quantitative experimental results demonstrate that cluster balanced sampling + selective querying outperform a selection of active learning methods in nearly all the tested datasets. The authors also show that their data-centric method can be integrated with model-centric methods to improve performance in the 1-shot setting.
- Some qualitative results of t-SNE visualizations of the class guided features vs image features alone show that the class guided features are better separated in the latent space. Gradient feature activation maps tend to support this with heatmaps showing a focus on classes of interest. Quantitative ablations confirm these observations.

**Audience:**

Yes

**Audience Explanation:**

There is a lot of interest in tuning VLMs to perform well in low-budget scenarios, and active learning with prompt tuning is certainly a promising approach. I think the proposed method as a whole is of broad interest with multiple experiments probing the efficacy of its individual components. Even though GradFAM is not directly relevant to model performance, it's still an interesting modification and might be of use in other contexts as well, say representation learning.

**Claims And Evidence:**

No

**Claims Explanation:**

The claims are mostly supported by the experimental results presented in the paper, but the baselines chosen for comparison are not the strongest, so I'm unsure if the results will hold up under more stringent conditions.

- **query methods:** CoreSet and BADGE are somewhat older methods and not particularly well suited to the low-budget AL regime. More modern methods like TypiClust [1], ProbCover [2], and DropQuery [3] might be more suitable for baselines, since all of them also address the cold start problem using some form of clustering. At this point it is known that any AL method not mitigating the cold start problem will suffer from diminished early performance, so it is unsurprising that the proposed method does well.
- **novelty of cluster balanced selection:** Modern AL methods (TypiClust, ALFA-mix [4], DropQuery) utilize cluster based sampling for diversity in their selections. TypiClust is probably the most similar in terms of partitioning the data points into clusters and then sampling from the densest regions (as opposed to centroid in this paper). It may be worth emphasizing that the proposed cluster based selection is novel in the context of active prompt tuning, but not necessarily as an AL method.
- **class guided features:** In the limit, I would expect the text features to align almost perfectly with the image features, so the computed soft label weights would approach zero for the incorrect classes and one for the correct class, so there would be little information added. While the current method shows quantitative improvements in table 2, it's unclear how much of that improvement would persist when using larger VLMs with better image-text alignment (and therefore approaching the aforementioned limit). Since current experiments were performed with ViT-B/32, it is worth investigating other model sizes like ViT-H/14 or even the giant variants of EVA-CLIP.

[1] [Active Learning on a Budget: Opposite Strategies Suit High and Low Budgets](https://arxiv.org/abs/2202.02794)

[2] [Active Learning Through a Covering Lens](https://arxiv.org/abs/2205.11320)

[3] [Revisiting Active Learning in the Era of Vision Foundation Models](https://arxiv.org/abs/2401.14555)

[4] [Active Learning by Feature Mixing](https://arxiv.org/abs/2203.07034)

**Requested Changes:**

I would like to see a few additional experiments to compare against stronger AL baselines and with larger VLMs.

- It would be good to add some of the modern methods mentioned previously to the results in table 3, specifically those addressing the cold start problem for a fairer comparison, since that seems to be where the majority of gains originate.
- As an ablation, it would be interesting to see the same experiments conducted with identical initial selections for CB+SQ to control for initialization differences, since centroid based initialization has been shown to improve performance across methods in [3].
- Repeating the comparison in table 2 with larger ViTs would help to determine if the class-guided features are truly beneficial compared to using just image features.

---

> ### Author Response · Authors · 2025-08-11
> **Response to Reviewer 9bb1 (Part 1/3)**
>
> ### **General Reply**
> We are grateful to the reviewer for your insightful and constructive feedback. **We will address the remaining concerns as soon as possible, as time permits.**
>
> ### **Weakness 1. Experiments with Larger VLMs**
>
> > Repeating the comparison in table 2 with larger ViTs would help to determine if the class-guided features are truly beneficial compared to using just image features.
>
> To address your concern, we repeat the Table 2 experiments on CLIP with a ViT-L/14-336 backbone [1] and on EVA01-CLIP-g-14-plus with a ViT-H/14 backbone [2]. The results in Tables A and B show that in both architecture the joint use of image and weighted text features outperforms using either alone.
>
> **Table A.** *Ablation studies on different feature spaces using CLIP with a ViT-L/14-336.* In the initial round, our class-guided features, leveraging both image and weighted text features, demonstrate effectiveness across 7 datasets.
> | Types | Caltech101 | OxfordPets | StanfordCars | Flowers102 | Food101 | FGVCAircraft | DTD | Avg. |
> |:---:|:---:|:---:|:---:|:---:|:---:|:---:|:---:|:---:|
> | Image Features | $83.17_{\pm 0.86}$ | $81.38_{\pm 0.30}$ | $57.53_{\pm 0.52}$ | $84.95_{\pm 0.94}$ | $76.02_{\pm 0.18}$ | $\mathbf{33.14_{\pm 0.24}}$ | $45.71_{\pm 1.24}$ | $65.99_{\pm 0.52}$ |
> | Weighted Text Features | $78.74_{\pm 2.13}$ | $78.13_{\pm 0.87}$ | $57.76_{\pm 0.78}$ | $81.89_{\pm 0.32}$ | $63.67_{\pm 0.73}$ | $28.53_{\pm 0.28}$ | $41.47_{\pm 1.64}$ | $61.46_{\pm 0.61}$ |
> | Class-guided Features (ours) | $\mathbf{85.07_{\pm 0.28}}$ | $\mathbf{82.52_{\pm 0.49}}$ | $\mathbf{63.08_{\pm 0.41}}$ | $\mathbf{88.62_{\pm 0.59}}$ | $\mathbf{81.06_{\pm 0.27}}$ | $\mathbf{33.14_{\pm 0.40}}$ | $\mathbf{46.33_{\pm 0.87}}$ | $\mathbf{68.55_{\pm 0.13}}$ |
>
> **Table B.** *Ablation studies on different feature spaces using EVA01-CLIP-g-14-plus with a ViT-H/14.* In the initial round, our class-guided features, leveraging both image and weighted text features, demonstrate effectiveness across 7 datasets.
> | Types | Caltech101 | OxfordPets | StanfordCars | Flowers102 | Food101 | FGVCAircraft | DTD | Avg. |
> |:---:|:---:|:---:|:---:|:---:|:---:|:---:|:---:|:---:|
> | Image Features | $79.48_{\pm 0.67}$ | $48.31_{\pm 0.64}$ | $46.85_{\pm 0.41}$ | $83.92_{\pm 0.23}$ | $62.84_{\pm 0.20}$ | $31.43_{\pm 0.61}$ | $\mathbf{44.15_{\pm 0.43}}$ | $56.71_{\pm 0.17}$ |
> | Weighted Text Features | $78.86_{\pm 0.77}$ | $40.89_{\pm 1.28}$ | $45.11_{\pm 0.17}$ | $81.02_{\pm 0.27}$ | $47.75_{\pm 0.03}$ | $29.07_{\pm 0.56}$ | $32.98_{\pm 0.29}$ | $50.81_{\pm 0.13}$ |
> | Class-guided Features (ours) | $\mathbf{81.08_{\pm 0.29}}$ | $\mathbf{57.43_{\pm 1.29}}$ | $\mathbf{52.97_{\pm 0.28}}$ | $\mathbf{87.84_{\pm 0.27}}$ | $\mathbf{69.06_{\pm 0.26}}$ | $\mathbf{31.47_{\pm 0.54}}$ | $42.28_{\pm 0.77}$ | $\mathbf{60.30_{\pm 0.32}}$ |
>
> > class guided features: In the limit, I would expect the text features to align almost perfectly with the image features, so the computed soft label weights would approach zero for the incorrect classes and one for the correct class, so there would be little information added. While the current method shows quantitative improvements in table 2, it's unclear how much of that improvement would persist when using larger VLMs with better image-text alignment (and therefore approaching the aforementioned limit). Since current experiments were performed with ViT-B/32, it is worth investigating other model sizes like ViT-H/14 or even the giant variants of EVA-CLIP.
>
> To address your question: in the case of a perfectly independent set of classes and an ideal model, as you mentioned, it would indeed be possible for the model to assign a value close to 1 to the single correct class for an image. However, such conditions are unlikely to occur in practice. For example, humans intuitively recognize that the semantic relationship between the classes “dog” and “wolf” is much closer than that between “dog” and “tree,” and we might even perceive an image as a wolf-like dog or a dog-like wolf. In this regard, I believe that soft labels offer a more realistic representation of real-world scenarios. Furthermore, while larger models may yield better performance and move closer to the ideal, in practice models such as CLIP with ViT-B/32 and ViT-H/14 achieve ImageNet zero-shot accuracies of only 72.8% and 78.0% [3], respectively, still a considerable distance from the ideal 100%.
>
> In addition, we conduct supplementary experiments on the aforementioned large backbone; details are provided in the third reviewer WqbA’s section, *Weakness 2. Generalization to Other Backbones*.
>
> **References**
> * [1] https://github.com/openai/CLIP/blob/main/clip/clip.py (As "ViT-L/14@336px" is available in Line 39, we experiment with this configuration.)
> * [2] https://github.com/baaivision/EVA/tree/master/EVA-CLIP
> * [3] https://github.com/mlfoundations/open_clip

---

> > ### Author Response · Authors · 2025-08-15
> > **Response to Reviewer 9bb1 (Part 2/3)**
> >
> > ### **Weakness 2. Clustering-based Baselines with Class-guided Features**
> >
> > > query methods: CoreSet and BADGE are somewhat older methods and not particularly well suited to the low-budget AL regime. More modern methods like TypiClust [1], ProbCover [2], and DropQuery [3] might be more suitable for baselines, since all of them also address the cold start problem using some form of clustering. At this point it is known that any AL method not mitigating the cold start problem will suffer from diminished early performance, so it is unsurprising that the proposed method does well.
> >
> > > It would be good to add some of the modern methods mentioned previously to the results in table 3, specifically those addressing the cold start problem for a fairer comparison, since that seems to be where the majority of gains originate.
> >
> > Thank you for suggesting several related approaches that address the cold-start problem through clustering. In Table C, we first evaluate the performance of these baselines in the initial round using only image features, finding that they alleviate the cold-start issue more effectively than Random selection. When class-guided features are added, all baselines benefit from a positive synergy that further enhances their performance. Specifically, Typiclust [1], ProbCover (K-means) [2], and DropQuery [3] all employ K-means clustering in the given feature space, selecting samples within each cluster based on typicality, coverage, or proximity to the centroid, respectively. Our method operates in the same manner as DropQuery [3] during initial pool selection, but differs in that we use class-guided features instead of image features. As shown in Table C and Figure 9(c), the proposed class-guided features can be easily integrated into existing baselines and yield substantial performance improvements.
> >
> > **Table C.** *Effect of class-guided features on initial pool selection.* In the initial round, incorporating class-guided features consistently improves baseline performance across six datasets.
> > | Type | Method | Caltech101 | OxfordPets | StanfordCars | Flowers102 | FGVCAircraft | DTD | Avg |
> > |:---|:---|:---:|:---:|:---:|:---:|:---:|:---:|:---:|
> > | | Random | $62.10_{\pm 0.42}$ | $46.95_{\pm 0.88}$ | $30.56_{\pm 0.54}$ | $53.36_{\pm 3.01}$ | $14.64_{\pm 0.37}$ | $25.35_{\pm 1.98}$ | $38.83_{\pm 0.65}$ |
> > | **Image Features** | TypiClust | $77.05_{\pm 0.20}$ | $58.27_{\pm 0.17}$ | $36.53_{\pm 0.15}$ | $76.64_{\pm 0.16}$ | $17.46_{\pm 0.49}$ | $41.11_{\pm 0.28}$ | $51.18_{\pm 0.11}$ |
> > | | ProbCover | $51.68_{\pm 0.23}$ | $53.02_{\pm 1.04}$ | $28.71_{\pm 0.16}$ | $51.12_{\pm 0.38}$ | $16.49_{\pm 0.34}$ | $32.76_{\pm 1.04}$ | $38.96_{\pm 0.15}$ |
> > | | ProbCover (K-means) | $75.34_{\pm 1.01}$ | $55.73_{\pm 0.49}$ | $34.36_{\pm 0.07}$ | $72.24_{\pm 1.34}$ | $16.55_{\pm 0.75}$ | $38.77_{\pm 2.59}$ | $48.83_{\pm 0.19}$ |
> > | | DropQuery | $77.97_{\pm 0.37}$ | $57.43_{\pm 0.33}$ | $37.39_{\pm 0.14}$ | $75.30_{\pm 0.48}$ | $17.09_{\pm 0.10}$ | $42.10_{\pm 0.34}$ | $51.21_{\pm 0.01}$ |
> > | **Class-guided Features** | TypiClust | $81.08_{\pm 0.27}$ | $71.45_{\pm 1.57}$ | $40.87_{\pm 0.36}$ | $78.83_{\pm 0.14}$ | $18.10_{\pm 0.17}$ | $40.05_{\pm 0.74}$ | $\mathbf{55.06_{\pm 0.06}}$ |
> > | | ProbCover | $63.95_{\pm 0.89}$ | $62.58_{\pm 1.91}$ | $32.61_{\pm 0.00}$ | $61.31_{\pm 0.32}$ | $15.81_{\pm 0.13}$ | $35.50_{\pm 0.51}$ | $\mathbf{46.99_{\pm 1.33}}$ |
> > | | ProbCover (K-means) | $78.86_{\pm 1.06}$ | $68.61_{\pm 1.67}$ | $39.11_{\pm 0.17}$ | $77.07_{\pm 0.37}$ | $17.83_{\pm 0.62}$ | $38.40_{\pm 0.77}$ | $\mathbf{53.31_{\pm 0.28}}$ |
> > | | DropQuery | $79.85_{\pm 0.32}$ | $70.95_{\pm 0.40}$ | $40.41_{\pm 0.53}$ | $77.86_{\pm 0.34}$ | $17.52_{\pm 0.36}$ | $39.11_{\pm 0.49}$ | $\mathbf{54.28_{\pm 0.10}}$ |
> >
> > **References**
> > * [1] Active Learning on a Budget: Opposite Strategies Suit High and Low Budgets, ICML 2022.
> > * [2] Active Learning Through a Covering Lens, NeurIPS 2022.
> > * [3] Revisiting Active Learning in the Era of Vision Foundation Models, TMLR 2024.

---

> > > ### Author Response · Authors · 2025-08-15
> > > **Response to Reviewer 9bb1 (Part 3/3)**
> > >
> > > ### **Weakness 3. Novelty of Cluster Balanced Selection**
> > >
> > > > novelty of cluster balanced selection: Modern AL methods (TypiClust, ALFA-mix [4], DropQuery) utilize cluster based sampling for diversity in their selections. TypiClust is probably the most similar in terms of partitioning the data points into clusters and then sampling from the densest regions (as opposed to centroid in this paper). It may be worth emphasizing that the proposed cluster based selection is novel in the context of active prompt tuning, but not necessarily as an AL method.
> > >
> > > We agree that cluster-based selection is well established in standard active learning [1, 2, 3], and we acknowledge that our proposed cluster-based selection is more valuable specifically in the active prompt tuning setting of VLMs, which use image and text encoders together. Importantly, we show that in VLMs such as CLIP, relying only on image features is suboptimal; incorporating text features alongside image features yields more informative queries and faster gains. This is supported by the experiments in Weakness 2 and further validated by the Coreset ablation in Figure 9(c), where including text features consistently improves selection effectiveness. We will revise the paper to clarify this scope of novelty and to better position our centroid-based, cluster-balanced strategy relative to prior sampling methods.
> > >
> > > ### **Weakness 4. Effect of Within-Cluster Selection Strategies on Class-Guided Features**
> > >
> > > > As an ablation, it would be interesting to see the same experiments conducted with identical initial selections for CB+SQ to control for initialization differences, since centroid based initialization has been shown to improve performance across methods in [3].
> > >
> > > Thank you for suggesting an interesting ablation study. We conduct experiments in which K-means clustering was performed on top of the class-guided features, followed by sample selection within each cluster using various strategies. Here, we note that we evaluate ProbCover [1], DropQuery [3], and Typiclust [2] exclusively as algorithms for initial pool selection.
> > >
> > > **Table D.** *Ablation study on different within-cluster selection strategies.*
> > > | Type | Method | Caltech101 | OxfordPets | StanfordCars | Flowers102 | FGVCAircraft | DTD | Avg |
> > > |:---|:---|:---:|:---:|:---:|:---:|:---:|:---:|:---:|
> > > | **Class-guided Features** | Random | $77.21_{\pm 1.56}$ | $63.87_{\pm 1.64}$ | $35.09_{\pm 0.84}$ | $65.19_{\pm 0.84}$ | $16.00_{\pm 0.60}$ | $35.76_{\pm 2.92}$ | $48.85_{\pm 0.75}$ |
> > > | | Entropy | $76.20_{\pm 0.53}$ | $41.11_{\pm 0.62}$ | $24.75_{\pm 0.11}$ | $50.87_{\pm 0.06}$ | $12.73_{\pm 0.23}$ | $30.04_{\pm 1.00}$ | $39.28_{\pm 0.20}$ |
> > > | | ProbCover | $78.86_{\pm 1.06}$ | $68.61_{\pm 1.67}$ | $39.11_{\pm 0.17}$ | $77.07_{\pm 0.37}$ | $17.83_{\pm 0.62}$ | $38.40_{\pm 0.77}$ | $53.31_{\pm 0.28}$ |
> > > | | DropQuery | $79.85_{\pm 0.32}$ | $70.95_{\pm 0.40}$ | $40.41_{\pm 0.53}$ | $77.86_{\pm 0.34}$ | $17.52_{\pm 0.36}$ | $39.11_{\pm 0.49}$ | $54.28_{\pm 0.10}$ |
> > > | | TypiClust | $\mathbf{81.08_{\pm 0.27}}$ | $\mathbf{71.45_{\pm 1.57}}$ | $\mathbf{40.87_{\pm 0.36}}$ | $\mathbf{78.83_{\pm 0.14}}$ | $\mathbf{18.10_{\pm 0.17}}$ | $\mathbf{40.05_{\pm 0.74}}$ | $\mathbf{55.06_{\pm 0.06}}$ |
> > >
> > > **References**
> > > * [1] Active Learning on a Budget: Opposite Strategies Suit High and Low Budgets, ICML 2022.
> > > * [2] Active Learning Through a Covering Lens, NeurIPS 2022.
> > > * [3] Revisiting Active Learning in the Era of Vision Foundation Models, TMLR 2024.

---

### Decision · Action_Editor_DkuN · 2025-09-27

**Recommendation:** Accept as is

**Audience:**

Yes

**Audience Explanation:**

Yes, there will be interest, as there is an active subcommunity working on this problem.

**Claims And Evidence:**

Yes

**Claims Explanation:**

Reviewers agreed that the method presented here for active prompt learning in vision-language models is new and its performance is sufficiently supported by empirical evidence.